# Indirect CRISPR screening with photoconversion revealed key factors of drug resistance with cell–cell interactions

Keisuke Sugita [1], Iichiroh Onishi[1], Ran Nakayama[1], Sachiko Ishibashi[1], Masumi Ikeda[1], Miori Inoue[1], Rina Narita[1], Shiori Oshima[1], Kaho Shimizu[1], Shinichiro Saito[1], Shingo Sato [2], Branden S. Moriarity[3], Kouhei Yamamoto[1], David A. Largaespada [3], Masanobu Kitagawa[1] & Morito Kurata [1✉]

Comprehensive screenings to clarify indirect cell–cell interactions, such as those in the tumor microenvironment, especially comprehensive assessments of supporting cells' effects, are challenging. Therefore, in this study, indirect CRISPR screening for drug resistance with cell–cell interactions was invented. The photoconvertible fluorescent protein Dendra2 was inducted to supporting cells and explored the drug resistance responsible factors of supporting cells with CRISPR screenings. Random mutated supporting cells co-cultured with leukemic cells induced drug resistance with cell–cell interactions. Supporting cells responsible for drug resistance were isolated with green-to-red photoconversion, and 39 candidate genes were identified. Knocking out C9orf89, MAGI2, MLPH, or RHBDD2 in supporting cells reduced the ratio of apoptosis of cancer cells. In addition, the low expression of RHBDD2 in supporting cells, specifically fibroblasts, of clinical pancreatic cancer showed a shortened prognosis, and a negative correlation with CXCL12 was observed. Indirect CRISPR screening was established to isolate the responsible elements of cell–cell interactions. This screening method could reveal unknown mechanisms in all kinds of cell–cell interactions by revealing live phenotype-inducible cells, and it could be a platform for discovering new targets of drugs for conventional chemotherapies.

[1] Department of Comprehensive Pathology, Graduate School of Medical and Dental Sciences, Tokyo Medical and Dental University (TMDU), Tokyo, Japan. [2] Center for Innovative Cancer Treatment, Tokyo Medical and Dental University (TMDU), Tokyo, Japan. [3] Masonic Cancer Center, University of Minnesota, Minneapolis, MN, USA. ✉email: kurata.pth2@tmd.ac.jp

Many oncogenes and tumor suppressor genes have been discovered, and anticancer drugs, including molecularly targeted drugs, are now widely used. However, cancer is still resistant to treatment, and drug resistance remains a major concern in discovering cures[1–3]. Therefore, the elucidation of drug resistance mechanisms is an important challenge to overcome. Regarding drug resistance systems, drug inactivation, drug target alteration, drug efflux, DNA damage repair, cell death inhibition, and the epithelial–mesenchymal transition are well-known and well-studied[4]. In addition to these mechanisms, the tumor microenvironment (TME), the peritumoral region composed of cancer-associated fibroblasts (CAFs), the extracellular matrix (ECM), immune cells, and vasculature have been the focus of research in recent years, as they co-evolve during malignant progression and contribute to cancer development, progression, and drug resistance[2,3,5].

The microenvironment was originally studied as part of the bone marrow as the bone marrow niche that maintains and regulates hematopoietic stem cells, and TME is also involved in the drug resistance of myeloma in the bone marrow, aside from solid tumors[6–8]. CAFs are also known to be stromal cells of the TME that have distinctive properties compared to fibroblasts derived from normal tissue and have been implicated in tumor growth, local invasion, distant metastasis, ECM remodeling, angiogenesis, and drug resistance[1,9–12]. Because of the wide variety of drug resistance mechanisms associated with cell–cell interactions in the TME, comprehensive analyses of these mechanisms are crucial for overcoming drug resistance.

According to global cancer statistics (GLOBOCAN 2020), pancreatic cancer has a poor prognosis, with the number of related deaths (466,003) being almost equal to the number of patients (495,773), and it is the seventh leading cause of cancer deaths in both men and women[13]. The 5-year survival rate is approximately 3–10%, and 80–85% of patients are ineligible for surgical resection at the time of diagnosis due to either the state of the locally advanced disease or distant metastases[14–17]. Pancreatic ductal adenocarcinoma (PDAC) is characterized by abundant desmoplastic stroma, and the TME in this stroma promotes drug resistance and a worse prognosis in PDAC[5,18,19].

In this study, an experimental model using CRISPR screening was created to more comprehensively elucidate the mechanism behind drug resistance induction by peritumoral supporting cells, whereas previous reports on TME or CAFs have mainly focused on TME tissue derived from cancers with poor prognosis[9,19–21].

In conventional CRISPR screening, the candidate genes for drug resistance are identified by detecting the increasing/decreasing number of tumor cells with guide RNAs (gRNAs) that induce random mutations under screening conditions, such as drug exposure[22]. However, in cases of drug resistance by cell–cell interactions from the surrounding environment, such as the TME, the identification of the responsible candidate genes for drug resistance with cell–cell interactions is difficult. This is because there is no selective increase in the number of surrounding cells, even if random mutations with gRNAs are induced in the surrounding cells (Fig. 1a). In this study, a CRISPR screening system, indirect CRISPR screening, using the photoconvertible fluorescent protein Dendra2 was established to identify responsible molecules in drug resistance induced by peritumoral cell–cell interactions.

## Results

**Indirect CRISPR screening with photoconversion**. In our experimental model, random mutations were inducted into supporting cells and co-cultured with the tumor cells to mimic the microenvironments of drug resistance and to identify the responsible cells using photoconversion and isolation. To identify and isolate cells that support drug resistance using the CRISPR library, we utilized a photoconversion protein, Dendra2, which can irreversibly convert its fluorescence wavelength from green to red with illumination from ultraviolet (UV) light (405 nm laser) (Supplementary Fig. 1)[23]. HEK293T and U937 leukemia cells, which were easy to handle technically, were inducted as cell–cell interaction systems to mimic a microenvironment. Dendra2-inducted HEK293T cells, with the CRISPR knockout library also inducted, were co-cultured with U937 cells (Fig. 1b, Step 1). U937 is relatively more sensitive to cytarabine than HEK293T (Supplementary Fig. 2a, b), and co-culture screening was performed with 3 μM of cytarabine which can kill most U937 cells to observe outstanding-alive U937 cells. The CRISPR library induction to HEK293T cells and screening were performed independently twice. The screenings were performed with $7.14 \times 10^6$ cells of HEK293T and $1.44 \times 10^7$ cells of U937 in 15 × 96-well plates per each screening. Indeed, most U937 cells were killed by cytarabine, actually, no viable colonies were observed in 12 control-wells with HEK293T cells without library induction. As we expected, a small number of U937 in close proximity to HEK293T survived and proliferated in a mulberry-like manner (Fig. 1b, Step 2). The survival U937 colonies in 96-well plates were identified as many as we can observe and resulted in 99 wells from the first screening and in 82 wells from the second screening. To obtain sufficient supporting cells for analysis and to confirm their ability to induce drug resistance reproducibly, all cells in the viable U937 colony-positive wells were transferred and re-seeded on six-well plates at 11 wells from the first screening and 8 wells from the second screening (Fig. 1b, Step 3). All supporting cells close to the viable U937 colonies under cytarabine exposure observed in the 19 wells were applied to photoconversion. The supporting cells that induced drug resistance near viable U937 colonies were identified by differential interference images from laser scanning microscopy, and photoconversion was performed with the 405 nm laser (Fig. 1b, Step 4). After photoconversion, the red form of Dendra2 was confirmed with laser scanning microscopy (559 nm laser) (Fig. 1b, Step 5) and the PI filter of FACS. Then, 1000 HEK293T cells with the red form of Dendra2 were sorted from each well by FACS (Fig. 1b, Step 6 and 1c).

Collected mutated HEK293T cells were then expanded and the mutation was analyzed using TA cloning, after which 39 candidate genes were obtained (Fig. 1b, Step 7 and Supplementary Table 1). Of the 39 genes, 21 were from the first screening, and 18 were from the second screening. These 39 candidate genes were validated to determine whether they could reproducibly induce drug resistance and associated mechanisms (Fig. 1d).

**Validation of leukemic drug resistance induced by supporting cells**. To validate the drug resistance functions of the 39 candidate genes in cell–cell interactions, each candidate gene in HEK293T was knocked out, and GFP-positive U937 cells were co-cultured to estimate the number of surviving tumor cells under cytarabine exposure using fluorescent microscopy (Fig. 2a). In the present co-culture system, GFP-positive U937 cells represent viable U937 cells (Supplementary Fig. 2d, e). gRNA-inducted HEK293T and GFP-positive U937 were co-cultured and exposed to 5 μM of cytarabine (Supplementary Fig. 2a, b) in 24-well plates for 48 h. The fluorescent images were evaluated, and the number of viable GFP-positive U937 cells was assessed (Fig. 2b and Supplementary Fig. 3). The results showed that the surviving number of viable U937 cells was significantly increased when co-cultured with HEK293T gRNA induced for the following 11 candidate genes: *ACTRT2, C9orf89, C19orf70, C21orf33, ENAM, MAGI2, MLPH, OXSM, RHBDD2, SLCO1B1*, and *ZNF48*.

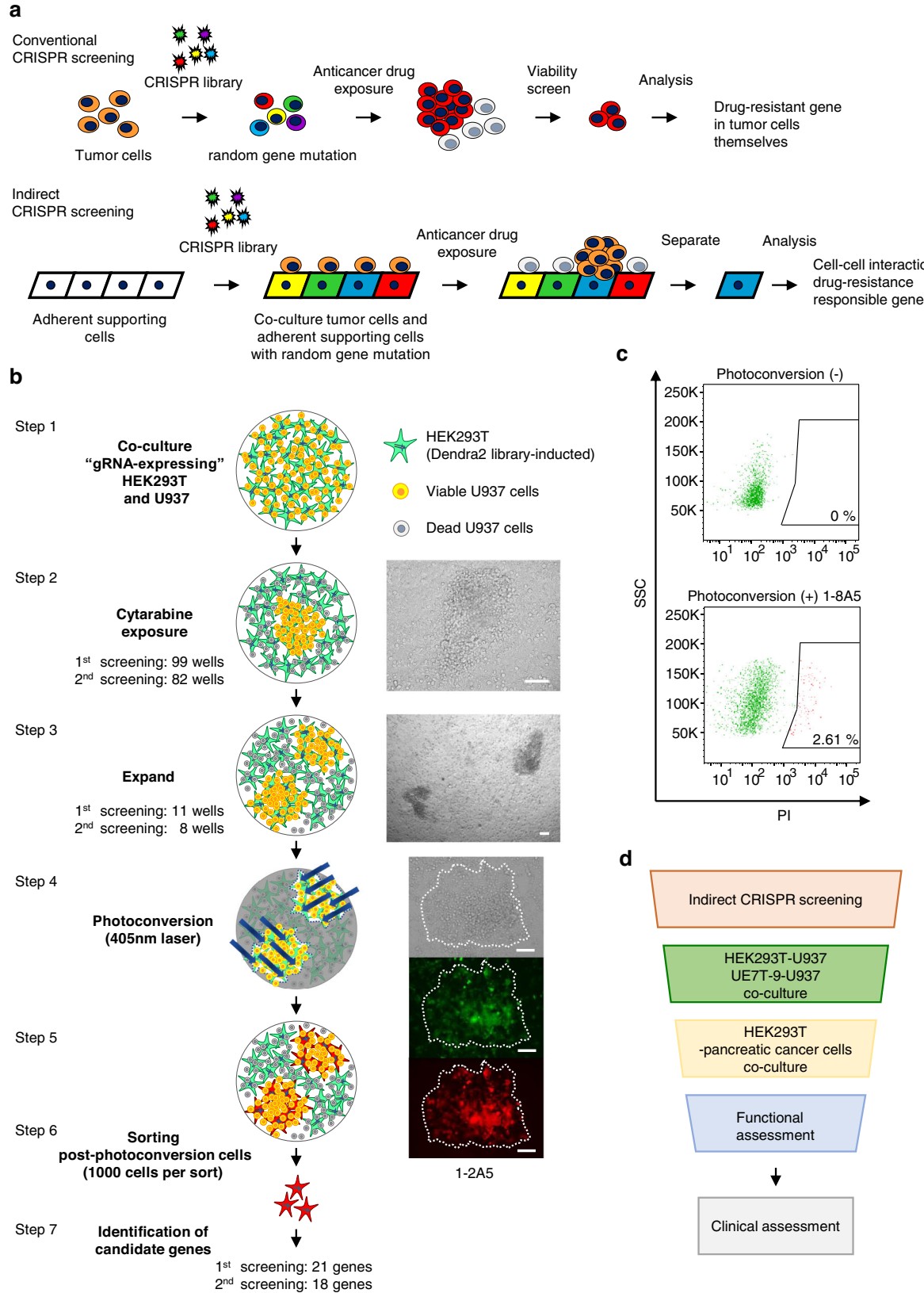

To confirm the universality (i.e., not specific in HEK293T cells) of the gene functions of these 11 candidates in cell–cell interactions, the human bone marrow-derived mesenchymal stem cell (UE7T-9) was used instead of HEK293T, which was used as an experimental model for cell–cell interactions. The same experiment was conducted to verify whether the function of drug resistance could be induced in the co-culture experiment with UE7T-9 and U937 (Fig. 2c and Supplementary Figs. 2b, c and 4). The results showed that the number of surviving GFP-positive U937 cells increased significantly with UE7T-9 by knocking out the 11 candidate genes of *ACTRT2, C9orf89, C19orf70, C21orf33, ENAM, MAGI2, MLPH, OXSM, RHBDD2,*

**Fig. 1 Indirect CRISPR screening system with photoconversion. a** The experimental model. In conventional CRISPR screening, the candidate genes for drug resistance are identified by detecting the increasing/decreasing number of tumor cells with gRNAs inducing random mutations under screening conditions (upper tier). Meanwhile, in indirect CRISPR screening, random mutations are inducted into the adherent supporting cells and co-cultured with the tumor cells to create so-called "microenvironments." The responsible supporting cells are then separated and analyzed for drug resistance with cell–cell interactions (lower tier). **b** Indirect CRISPR screening system with Dendra2. HEK293T and U937 cells were inducted as cell–cell interaction systems to mimic the microenvironment. U937 was co-cultured with HEK293T inducted with Dendra2 and the CRISPR library (Step 1). Under cytarabine exposure, most U937 cells were killed, and only some U937 cells close to HEK293T survived and proliferated (Step 2). The supporting cells that induced drug resistance in tumor cells were expanded (Step 3) and then identified by laser scanning microscopy. They then underwent photoconversion with the 405 nm laser (Steps 4 and 5). After photoconversion, the red-fluorescing HEK293T cells were sorted on FACS (Step 6). Sorted cells were analyzed to identify the genes responsible for drug resistance induced by cell–cell interactions (Step 7). Microscopic images are representative images (from the well "1–2A5") corresponding to each schema (bright-field image and laser scanning images; green: 473 nm, red: 559 nm). Scale bar, 50 μm. **c** FACS sorted supporting cells without (upper tier) or with (lower tier) photoconversion using the PI filter. Photoconverted cells were sorted by 1000 cells per well. **d** Pipeline of validation experiment.

*SLCO1B1*, and *ZNF48*, respectively. Therefore, the above 11 candidate genes could induce universal drug resistance with cell–cell interactions and were explored further.

**Application to pancreatic models of drug resistance with cell–cell interactions.** Pancreatic cancer is associated with abundant stroma, and this peritumoral stroma is associated with drug resistance[18]. Additionally, cytarabine and gemcitabine are both pyrimidine analogues and have similar chemical structures[24]. Therefore, candidate genes that induced cytarabine resistance were also evaluated to determine whether they induced gemcitabine resistance. The candidate genes that induced drug resistance in U937 cells in co-culture with HEK293T or UE7T-9 cells were applied to assess whether they universally induce drug resistance with cell–cell interactions in other cancers. Invasive pancreatic ductal carcinoma cell lines MIA PaCa-2 and SUIT-2 with HEK293T and the anticancer drug gemcitabine were conducted in co-culture experiments.

As the sensitivity among HEK293T, MIA PaCa-2, and SUIT-2 to gemcitabine was comparable (IC50s against gemcitabine were as follows; HEK293T: $1.46 \times 10^{-3}$ μM, MIA PaCa-2: $4.10 \times 10^{-2}$ μM, SUIT-2: $8.33 \times 10^{-3}$ μM, Supplementary Fig. 5a–c), *Deoxycytidine kinase* (*DCK*) knockout HEK293T was generated. DCK deficiency is known to be a major factor in gemcitabine resistance in vitro and in vivo due to its central role in gemcitabine metabolism[25]. Cloning of knockout HEK293T cells was performed using limiting dilution methods, and CRISPR knockout clones (CKO) were established (Supplementary Fig. 5d). IC50 against gemcitabine was remarkably increased in HEK293T *DCK*-CKO (23.8 μM, Supplementary Fig. 5e) compared to MIA PaCa-2 or SUIT-2. To differentiate between HEK293T and pancreatic cancer cell lines under fluorescence microscopy, only pancreatic cancer cells were transfected with GFP for co-culture and gemcitabine exposure experiments (Supplementary Fig. 5f).

The number of surviving GFP-positive pancreatic cancer cells under gemcitabine exposure increased significantly universally in MIA PaCa-2 (Fig. 3a and Supplementary Fig. 6a) and SUIT-2 (Fig. 3b and Supplementary Fig. 6b) when co-cultured with HEK293T, which was knocked out for eight candidate genes: *C9orf89, C19orf70, C21orf33, ENAM, MAGI2, MLPH, RHBDD2,* and *SLCO1B1*. The knockout of these eight genes in the supporting cells universally induced the survival of leukemia cells under cytarabine and pancreatic cancer cells under gemcitabine.

**Addressing biologically responsible factors in drug resistance with cell–cell interactions.** To assess the universal and obvious effects of drug resistance with cell–cell interactions, highly expressed candidate genes in HEK293T, specifically *C9orf89, C19orf70, C21orf33, MAGI2, MLPH,* and *RHBDD2* (Supplementary Fig. 7),

were focused on and applied to the next experiments. CKOs of HEK293T were established for six candidates (Supplementary Figs. 8a–d and 9a, b) and co-cultured with U937, and cytarabine-induced apoptosis was assessed using cleaved caspase-3 (Fig. 4a). In HEK293T alone under cytarabine, very few cleaved caspase-3-positive cells were observed. In the co-culture system, the significant inhibition of apoptosis was observed in the group co-cultured with HEK293T cells of *C9orf89*-CKO, *MAGI2*-CKO, *MLPH*-CKO, or *RHBDD2*-CKO compared to the control. Furthermore, no significant differences in the viabilities of HEK293T cells of *C9orf89*-CKO, *MAGI2*-CKO, *MLPH*-CKO, *RHBDD2*-CKO, and Non-target were observed under cytarabine exposure (Supplementary Fig. 10).

To investigate the biological features of *C9orf89*, *MAGI2*, *MLPH*, and *RHBDD2*, RNA sequences were performed on HEK293T cells of *C9orf89*-CKO, *MAGI2*-CKO, *MLPH*-CKO, and *RHBDD2*-CKO by next-generation sequencers. In gene set enrichment analysis (GSEA), 66 categories were universally upregulated in HEK293T cells of *C9orf89*-CKO (Supplementary Data 1a), *MAGI2*-CKO (Supplementary Data 1b), *MLPH*-CKO (Supplementary Data 1c), and *RHBDD2*-CKO (Supplementary Data 1d). In these 66 categories (Supplementary Data 1e), two categories that had been suggested to be related in drug resistance induced in the microenvironment based on previous studies were "cytokine activity" (Fig. 4b) and "cell adhesion mediated by integrin"[1,10,12]. In these two categories, *CXCL12* in "cytokine activity" was the most commonly enriched in the four HEK293T-CKO clones based on the rank metric score in GSEA (Supplementary Data 1f, g). *CXCL12* was commonly upregulated in HEK293T cells of *C9orf89*-CKO, *MAGI2*-CKO, *MLPH*-CKO, and *RHBDD2*-CKO (Fig. 4c), therefore was focused on in subsequent experiments. To confirm the expression of *CXCL12* in CKO clones, qPCR was performed. *CXCL12* expression was significantly upregulated in HEK293T cells of *C9orf89*-CKO, *MAGI2*-CKO, and *RHBDD2*-CKO (Fig. 4d). To confirm the cell–cell universality of elevated *CXCL12* expression, *CXCL12* expression was examined by qPCR in UE7T-9 cells with CKO mutations of these four candidates (Supplementary Figs. 11a–c and 12). UE7T-9 *RHBDD2*-CKO showed elevated *CXCL12* expression, while UE7T-9 cells of *C9orf89*-CKO, *MAGI2*-CKO, and *MLPH*-CKO showed no upregulation of *CXCL12* (Fig. 4e). Secretion levels of CXCL12 were upregulated universally in *C9orf89*-CKO, *MAGI2*-CKO, *MLPH*-CKO, and *RHBDD2*-CKO clones of HEK293T and UE7T-9 cells (Fig. 4f, g).

These results confirm that *RHBDD2*-CKO might upregulate *CXCL12* expression in any type of stromal cell.

**RHBDD2 regulates CXCL12 expression in supporting cells and CXCL12 induces anticancer drug resistance via the PI3k-Akt-mTOR pathway.** To verify that the cytarabine resistance of U937 is induced by the secretion of *RHBDD2*-CKO HEK293T, U937

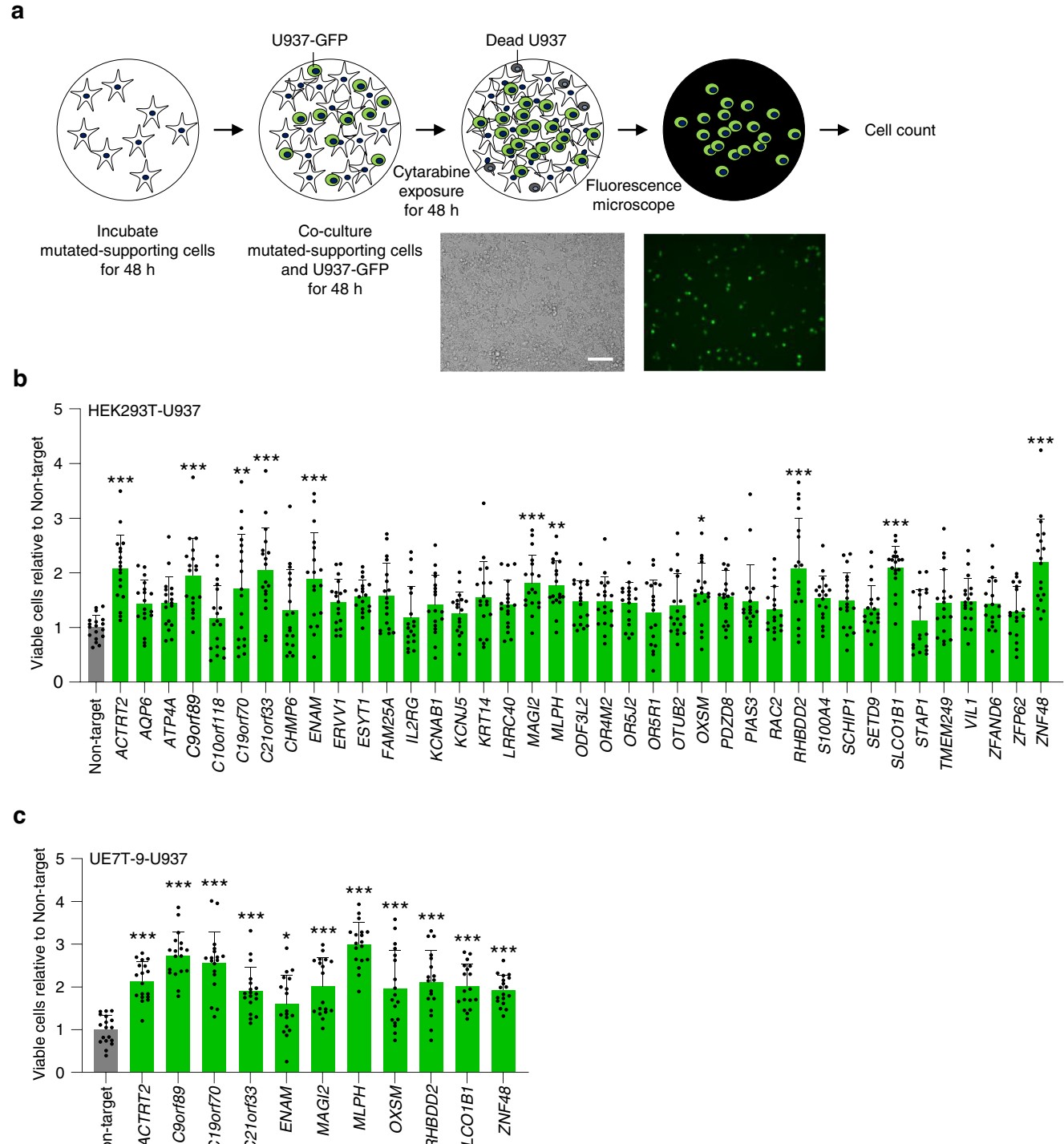

**Fig. 2 Validation of drug resistance with cell–cell interactions in HEK293T-U937 and UE7T-9-U937 model. a** Experimental scheme for the detection of living U937 cells in co-culture experiments. Scale bar, 50 μm. **b**, **c** The cell count of viable U937 cells co-cultured with knockout mutant-supporting cells (HEK293T (**b**) and UE7T-9 (**c**)) treated with 5 μM of cytarabine for 48 h. Three fields (×20 objective field) were randomly captured for each well, and the numbers of viable U937 cells were counted. The *x* axis represents the knockout candidate genes of the supporting cells. The *y* axis represents the number of viable cells relative to that of co-cultured cells with the control (non-targeted gRNA) of supporting cells. All experiments were performed in biological triplicate in each of the two independent experiments. Data are represented as mean ± SD. Statistical significance values were calculated by performing one-way ANOVA using Dunnett's test (**b**, **c**). *$P < 0.05$; **$P < 0.01$, and ***$P < 0.001$.

was cultured with supernatant from *RHBDD2*-CKO HEK293T and exposed to 250 nM of cytarabine. The viability of U937 cultured with the *RHBDD2*-CKO culture supernatant was slightly but significantly increased under cytarabine exposure (Supplementary Fig. 13).

To verify that cytarabine resistance of U937 induced by the co-culture with *RHBDD2*-CKO HEK293T or *RHBDD2*-CKO UE7T-9 is mediated by CXCL12, cytarabine exposure experiments were also performed after pre-treatment with the CXCL12-neutralizing antibody. The results showed that cytarabine exposure following

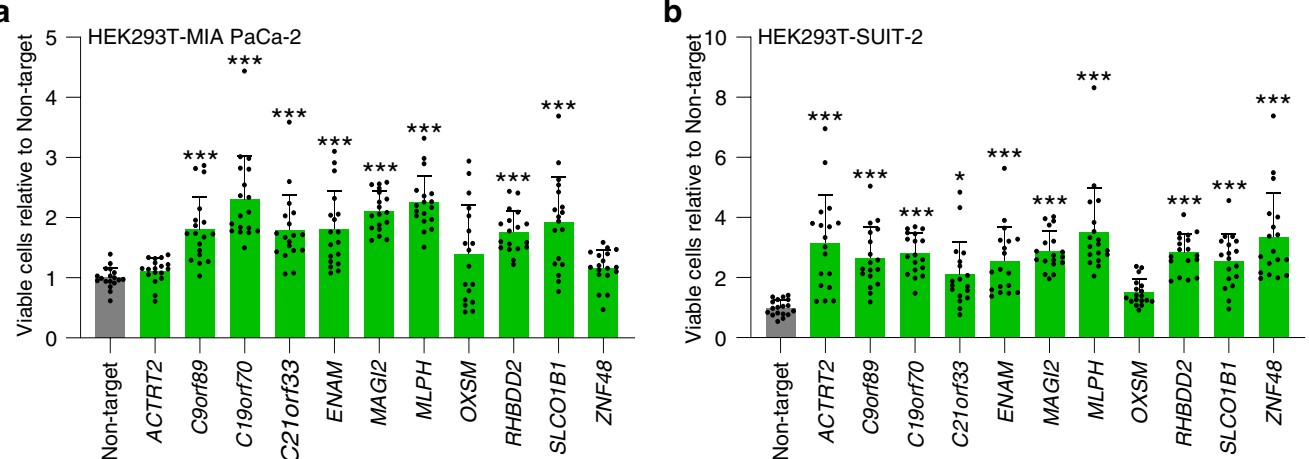

**Fig. 3 Validation of drug resistance with cell–cell interactions in HEK293T-pancreatic cancer cells model. a, b** The cell counts of viable pancreatic cancer cells (MIA PaCa-2 (**a**) and SUIT-2 (**b**)) co-cultured with knockout mutant-HEK293T treated with gemcitabine (10 µM (**a**) or 3 µM (**b**)) for 48 h. Three fields (×20 objective field) were randomly captured for each well, and the numbers of viable pancreatic cancer cells were counted. The x axis represents the knockout candidate genes of HEK293T. The y axis represents the number of viable cells relative to that co-cultured with the control (non-targeted gRNA) of HEK293T. All experiments were performed in biological triplicate in each of the two independent experiments. Data are represented as mean ± SD. Statistical significance values were calculated by performing one-way ANOVA using Dunnett's test (**a**, **b**). *P < 0.05 and ***P < 0.001.

the treatment with CXCL12-neutralizing antibodies increased cleaved caspase-3-positive cells under cytarabine exposure, which was similar to that observed when the cells were co-cultured with control HEK293T or UE7T-9 (Fig. 4h, i). Co-culture experiments of parental HEK293T-U937 cells and parental UE7T-9-U937 cells with cytarabine exposure after pre-treatment with the recombinant CXCL12 were also performed. The results showed that cytarabine exposure following the treatment with recombinant CXCL12 decreased cleaved caspase-3-positive cells under cytarabine exposure, which was similar to the effect observed when the cells were co-cultured with HEK293T RHBDD2-CKO or UE7T-9 RHBDD2-CKO (Supplementary Fig. 14a, b). In addition, in the U937 mono-culture experiment, cleaved caspase-3-positive cells under cytarabine exposure were slightly but significantly decreased (Supplementary Fig. 14c). These results suggest that a high concentration of CXCL12 secreted from the supporting cells might induce cytarabine resistance in U937.

A known pathway for the induction of CXCL12-mediated anticancer drug resistance is the activation of the PI3K-Akt-mTOR system in tumor cells[12]. Therefore, whether elevated CXCL12 in RHBDD2-CKO HEK293T or RHBDD2-CKO UE7T-9 results in the activation of the PI3K-Akt-mTOR system in U937 was tested. Phosphorylation of Akt in U937 cells under co-culture with HEK293T or UE7T-9 cells was evaluated. Increased phosphorylation of Akt in U937 (labeled with GFP to differentiate from HEK293T or UE7T-9) cells co-cultured with HEK293T or UE7T-9 was assessed by flow cytometry. The results showed that phosphorylated-Akt (phospho-Akt) was increased in U937 co-cultured with RHBDD2-CKO HEK293T or RHBDD2-CKO UE7T-9 (Fig. 4j, k and Supplementary Fig. 15).

**Analysis of the expression of cell–cell interaction factors in clinical pancreatic cancers.** To evaluate whether four important candidate genes in stromal cells, C9orf89, MAGI2, MLPH, and RHBDD2, are associated with prognosis, immunohistochemical staining of C9orf89, MAGI2, MLPH, or RHBDD2 in pancreatic ductal carcinoma surgical resection samples without neoadjuvant chemotherapy (n = 60) was conducted (Supplementary Data 2). Immunostainability of fibroblasts surrounding pancreatic carcinoma cells was evaluated. The stainabilities of the cells in the peritumoral stroma were scored as follows: not stained in

fibroblasts was (−), positive images in only a few fibroblasts or weakly positive images were (1 +), and strongly positive on most of the fibroblasts was (2 +). Based on scoring, (−) and (1 +) were classified as negative groups, and (2 +) was the positive group (Fig. 5a and Supplementary Fig. 16). Overall survival (OS) was significantly shortened in the negative group compared to the positive group that expressed MAGI2 and RHBDD2 (MAGI2: P = 0.014, RHBDD2: P < 0.001) (Fig. 5b and Supplementary Table 2). However, for the MLPH and C9orf89 groups, no significant correlation between immunostainability and OS could be identified (MLPH: P = 0.514, C9orf89: P = 0.089). Six parameters, including age, pT category, tumor size, UICC stage, MAGI2, and RHBDD2, were tested by multivariate analysis using the Cox proportional hazard model (Supplementary Table 2). The results showed that the pT category was 3–4 (vs. 1–2, hazard ratio: 8.665, P = 0.013), UICC stage was III–IV (vs. I–IIB, hazard ratio: 3.051, P = 0.003), and the RHBDD2-negative group (vs. RHBDD2-positive group, hazard ratio: 13.590, P < 0.001) independently predicted shorter OS.

Additionally, immunohistochemical staining of CXCL12 was conducted to assess the association between RHBDD2 and CXCL12 expression in fibroblasts surrounding carcinoma cells. As a result, an inverse correlation was found between RHBDD2 and CXCL12 expression (Fig. 5c).

**Discussion**
In this study, Dendra2, which can convert its fluorescence wavelength by UV laser illumination, was introduced into adherent supporting cells combined with the CRISPR library and succeeded in isolating drug resistance inducible supporting cells close to tumor cells surviving and proliferating under anticancer drug exposure. This gene screening system, which focuses on labeling and isolating peritumoral responsible mutant cells, not selectively growing tumor cells, under microscopic observation, has not been reported previously and is a useful technique in comprehensive screenings of cell–cell interactions. However, it would be more efficient to search for candidate genes if single-cell RNA-seq was conducted for photoconverted cell clones without the expansion of targeted cells and the automation of laser illumination on the targeted cells under the fluorescence microscope.

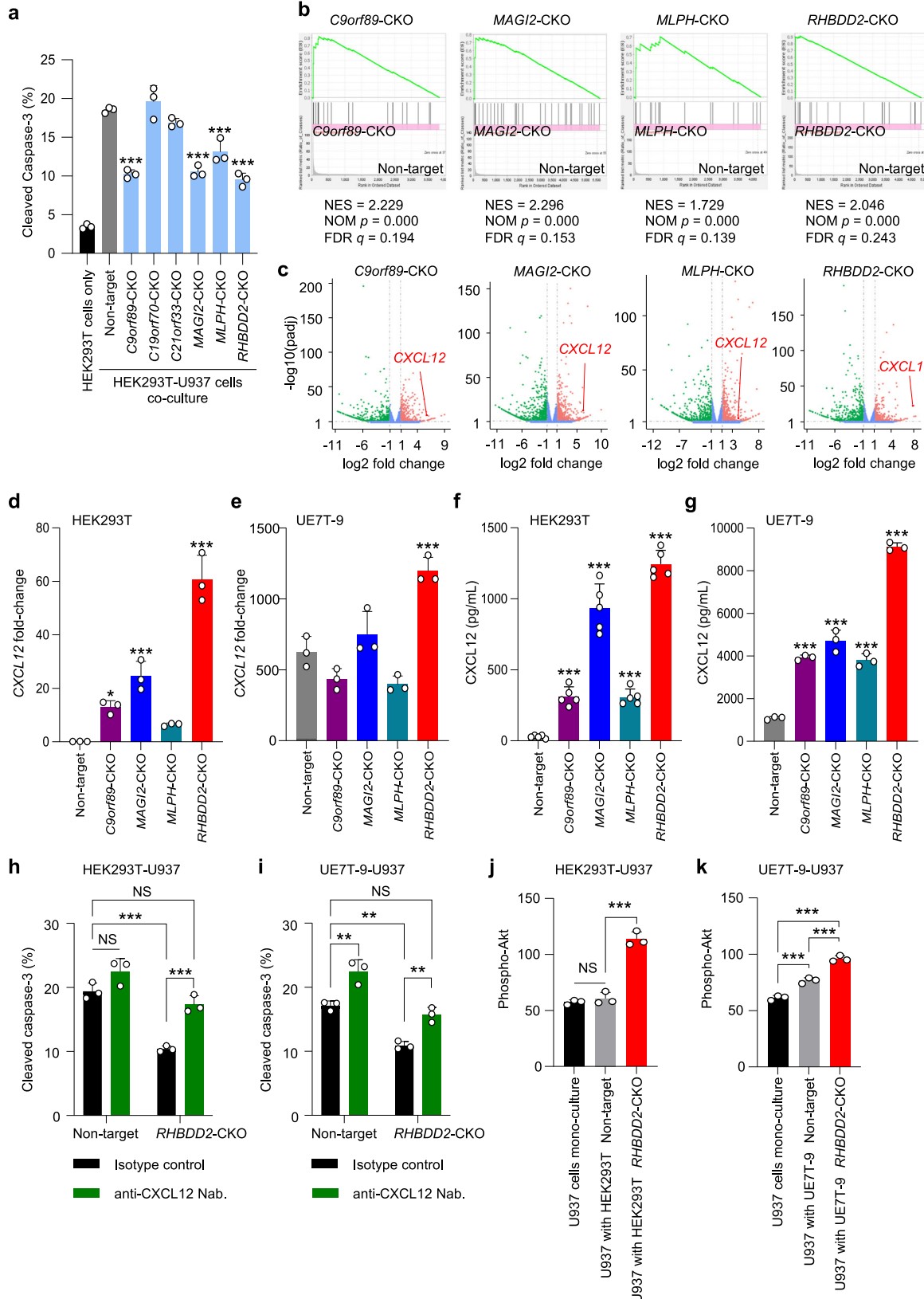

In the present study, there are 39 candidate genes identified by screening. However, only 11 genes were successfully validated with drug resistance functions. There could be many reasons why some candidate genes were identified by screening but not successfully validated. Generally, this will have been caused by the technical limitations of random screening and the original systems used in the present study. In the screening experiment, HEK293T cells, in close proximity to viable colonies identified under microscopic observation, were manually illuminated by the laser. We speculate that the false-positive rate in labeling candidate cells was caused by the inevitable exposure of the laser beam to non-objective cells surrounding the objective candidate cells.

**Fig. 4 Addressing biologically responsible factors in drug resistance through cell–cell interactions with RNA-seq and the regulation of *CXCL12* in *RHBDD2*-CKO cells. a** Ratio of apoptotic cells in the co-culture system with *C9orf89*-CKO, *C19orf70*-CKO, *C21orf33*-CKO, *MAGI2*-CKO, *MLPH*-CKO, and *RHBDD2*-CKO HEK293T under cytarabine exposure for 48 h. **b** GSEA of HEK293T-CKO clones vs. HEK293T control showing enrichment of gene sets involved in cytokine activity. **c** Volcano plot of HEK293T-CKO clones vs HEK293T control showing the upregulation of *CXCL12* in *C9orf89*-CKO, *MAGI2*-CKO, *MLPH*-CKO, and *RHBDD2*-CKO HEK293T. **d** The mRNA expression of *CXCL12* was significantly upregulated in HEK293T *C9orf89*-CKO, *MAGI2*-CKO, *MLPH*-CKO, and *RHBDD2*-CKO. **e** The mRNA expression of *CXCL12* expression was significantly upregulated in UE7T-9 *RHBDD2*-CKO.
**f, g** CXCL12 secretion was significantly upregulated in HEK293T (**f**) and UE7T-9 (**g**) cells of *C9orf89*-CKO, *MAGI2*-CKO, *MLPH*-CKO, and *RHBDD2*-CKO. **h, i** In the co-culture experiment of U937 with *RHBDD2*-CKO HEK293T (**h**) or *RHBDD2*-CKO UE7T-9 (**i**), pre-treatment with CXCL12-neutralizing antibodies increased cleaved caspase-3-positive cells under cytarabine exposure for 48 h, which was similar to the effect observed when the cells were co-cultured with control HEK293T or UE7T-9. **j, k** Evaluation of phospho-Akt downstream of CXCL12 in GFP-positive U937 under mono- or co-culture conditions. Phospho-Akt was increased in U937 co-cultured with *RHBDD2*-CKO HEK293T (**j**) and *RHBDD2*-CKO UE7T-9 (**k**) for 48 h. Experiments were performed with biological triplication (**a**, **d**, **e**, **g–k**) or quintuplication (**f**) in three independent repeats. Data are represented as mean ± SD. Statistical significance values were calculated by performing one-way ANOVA with Dunnett's test (**a**, **d–g**) or one-way ANOVA with Bonferroni's test (**h–k**). *$P < 0.05$; **$P < 0.01$; ***$P < 0.001$; and NS: non-significant.

Additionally, in the co-culture validation experiments, the concentration of cytarabine was 5 μM, higher than the concentration used in screening, 3 μM, to select candidate genes of stronger phenotypes inducing anticancer drug resistance. Of course, off-target effects of gRNAs were also potentially responsible for false-positive candidates. To overcome the problem of false positives, repeated validation experiments were conducted using multiple cell types in the present study. We are currently considering methods to improve the system to isolate objective candidate cells accurately and efficiently.

*RHBDD2* was identified as a gene responsible for drug resistance with cell–cell interactions, and clinical data also supported that the loss of RHBDD2 in supporting cells was suggested as a poor prognosis factor. *RHBDD2* is a member of the rhomboid family of membrane-bound proteases and is known to be overexpressed in the advanced stages of breast cancers and colorectal cancers[26–28]. RHBDD2 is also reported that its overexpression promotes chemoresistance and invasive phenotypes of rectal cancer[28]. However, these previous studies focused on the expression of RHBDD2 in the tumor cells themselves, and there are no reports examining its expression in peritumoral supporting cells. The pathway linking RHBDD2 and CXCL12 is a finding in the present study, and the axis of RHBDD2–CXCL12 in cell–cell interactions could be a drug target (Fig. 5d).

The drug resistance induced by the TME includes the prevention of drug absorption and the immune clearance of tumor cells[2]. Cell–cell adhesion systems in stromal cells, ECM, and cancer cells, such as integrin αvβ1 interactions, are known to convey such signals[4]. Activating the tumor via secretory systems, such as the paracrine, juxtacrine, and autocrine, are also a mechanism by which cell–cell interactions in the microenvironment induce anticancer drug resistance, and CAF-derived secretion, such as through cytokines, chemokines, and growth factors, are known to be involved in tumor growth, progression, and drug resistance[1,3,11,12,20]. It is also known that PI3K-AKT-mTOR signaling pathway activation via CAF-derived secretion, such as through CXCL12, VCAM1, IL-22, CXCL5, HGF, and SPARC, induces tumor proliferation, migration, and stemness[12]. In pancreatic cancer, elevated CXCL12 in CAFs is known to promote tumor progression via CXCL12–CXCR4 interaction[29]. In the present screening experiment, the cytokine signaling pathway in GSEA was upregulated. *RHBDD2*-CKO-induced upregulation of CXCL12 in stromal cells leads to anticancer drug resistance via activating the PI3K-Akt-mTOR pathway in tumor cells.

In recent years, many genetic screening methods that combine perturbation screening based on the CRISPR libraries with single-cell analysis, such as Perturb-seq, CITE-seq, and Perturb-CITE-seq, have been established, and they have led to the development of studies into the relationship between mutation and function on a single-cell basis[30–34]. The characteristics of tumor cells with random mutations and their surrounding peritumoral stromal cells and inflammatory cells have been elucidated by applying spatial gene expression analysis technology, Visium, with CRISPR screening; however, this system screened the random mutation in the tumor cells and analyzed the expression profiles of stromal cells, i.e. not screened with random mutation in the stromal cells[35]. These techniques can analyze single supporting cells with randomly mutated tumors with CRISPR screening; however, it is difficult to isolate living supporting cells based on the distance and position of the tumors.

A screening method that combines the CRISPR library and photoconversion into cell–cell interactions was established. In future applications, CAFs/tumor-infiltrating lymphocytes (TILs) in actual tissues would be promising targets. This method could be useful to reveal unknown mechanisms behind all kinds of cell–cell interactions and collecting living cells is also a great advantage of biological analysis. It could also be a platform for discovering the new targets of drugs in combination with conventional chemotherapy.

## Methods

**Cell culture and transduction.** HEK293T, UE7T-9, MIA PaCa-2, and SUIT-2 cells were obtained from the JCRB Cell Bank (National Institutes of Biomedical Innovation, Health and Nutrition, Japan). The authentication of U937 cells was confirmed using a 10-loci multiplex short tandem repeat analysis provided by the cell authentication services of BEX Co., LTD (Japan). Other cells were used for experiments immediately after obtain. U937 tested negative for mycoplasma contamination and was frozen as aliquots and each vial was used for experiments immediately after thawing. All other cell lines were purchased by the vendor and immediately frozen as aliquots and each vial was used for the experiments immediately after thawing. Therefore, the mycoplasma contamination test is not performed. HEK293T and UE7T-9 cells were maintained in Dulbecco's modified Eagle's medium (DMEM, FUJIFILM Wako Pure Chemical, Japan), U937 and SUIT-2 cells were kept in Roswell Park Memorial Institute 1640 (RPMI, FUJIFILM Wako Pure Chemical), and MIA PaCa-2 cells were kept in Eagle's minimum essential medium (EMEM, FUJIFILM Wako Pure Chemical), respectively, with 10% fetal bovine serum (Thermo Fisher Scientific, MA, USA) and 1% penicillin/streptomycin (FUJIFILM Wako Pure Chemical) with 5% $CO_2$ at 37 °C. Regarding the transduction of the lentiviruses, $5 \times 10^5$ cells/well of HEK293T cells were seeded in a six-well plate one day before the transfection. The next day, 3 μg of lentivirus plasmid was transfected with 1 μg of pMD2.G and 2 μg of pCMV using the Lipofectamine 3000 reagent (Invitrogen, MA, USA). Twelve hours after transfection, the medium was changed to fresh DMEM. The virus supernatant was harvested at 48 h after transfection and then filtered with Millex-HP 0.45 μm (Millipore, MA, USA). Then, $1 \times 10^6$ cells/well of viral inducible cells were seeded in a six-well plate one day prior to transduction and were transduced with this lentiviral supernatant with 5 μg/ml polybrene (Sigma-Aldrich, MO, USA). Transduction was performed using spin infection followed by 30 min of centrifugation at 1800 rpm and additional incubation for 2 h. Then, the supernatant was changed for the fresh culture medium. After a 2-week selection with the antibiotics corresponding to resistance for each plasmid, cells were conducted in the following experiments.

HEK293T clones of *C9orf89*-CKO, *C19orf70*-CKO, *C21orf33*-CKO, *MAGI2*-CKO, *MLPH*-CKO, *RHBDD2*-CKO, and Non-target and UE7T-9 clones of

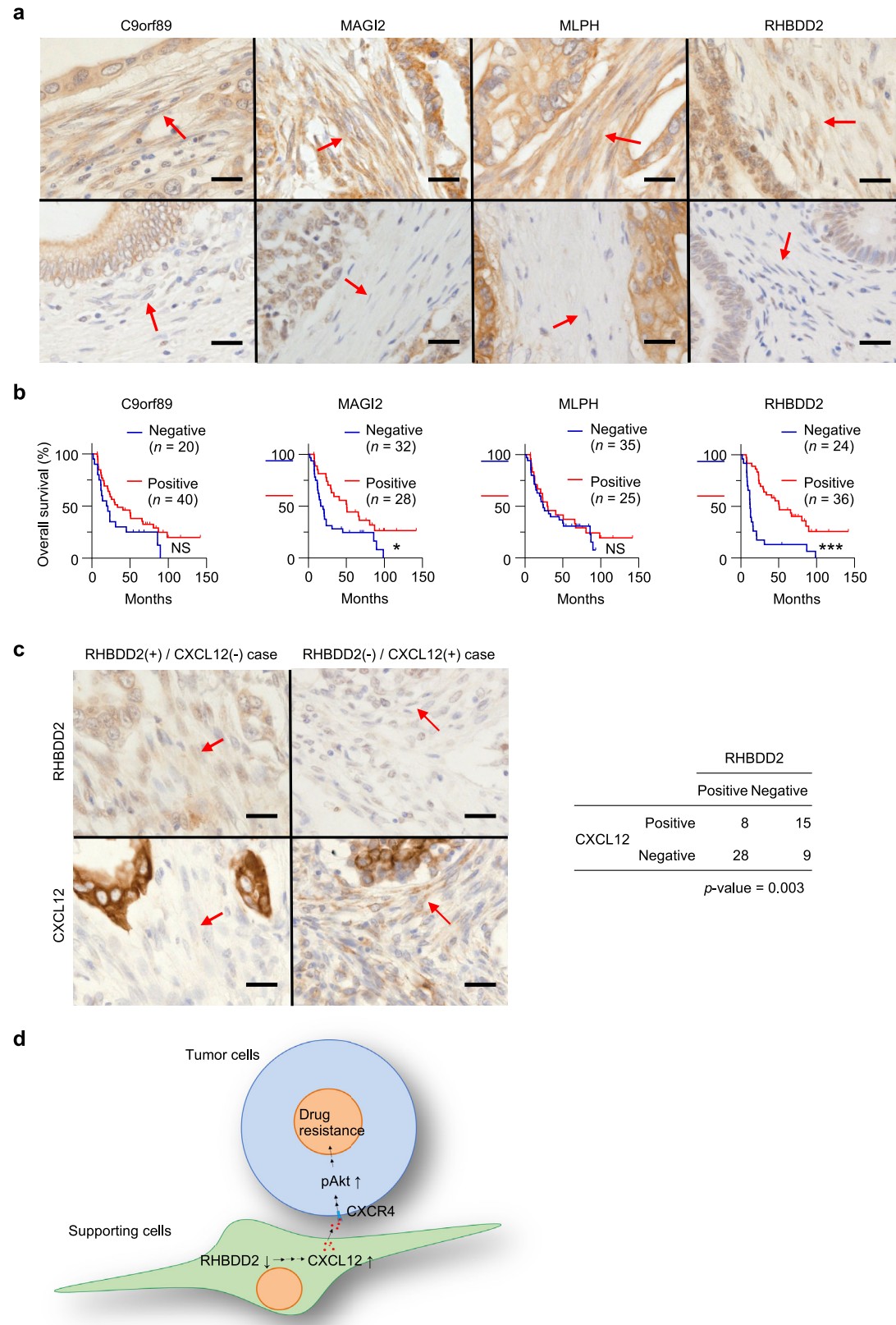

*C9orf89*-CKO, *MAGI2*-CKO, *MLPH*-CKO, *RHBDD2*-CKO, and Non-target were established with the limiting dilution methods.

All plasmids used in all experiments are listed in Supplementary Table 3.

**Indirect CRISPR screening**. A vector pDendra2-Hygro was generated by integrating the hygromycin-resistant gene and P2A sequence into the multi-cloning site at the N-terminus of the pDendra2-N Vector (Takara-Clontech, Japan). The lentiCas9-Blast was transduced to HEK293T cells and treated with blasticidin (10 µg/ml, InvivoGen, CA, USA) for 2 weeks. The pDendra2-Hygro was induced by lipofection in Cas9-expressing HEK293T cells and treated with hygromycin B (200 µg/ml, Invitrogen) for 2 weeks, and green fluorescence-positive cells were sorted using flow cytometry to purify the percentage of Dendra2-positive cells (MoFlo XDP, Beckman Coulter, CA, USA). The Human CRISPR Knockout Pooled

**Fig. 5 The expression of C9orf89, MAGI2, MLPH, and RHBDD2 in fibroblasts of human pancreatic cancers and their significance as prognostic factors.**
**a** Immunohistochemical staining of C9orf89, MAGI2, MLPH, or RHBDD2 in pancreatic ductal carcinoma surgical resection samples without neoadjuvant chemotherapy ($n = 60$). The immunostainability of fibroblasts surrounding pancreatic carcinoma cells was evaluated. Representative positive cases (upper tier) and negative cases (lower tier). **b** Kaplan–Meier survival curves of pancreatic ductal carcinoma patients with negative and positive groups for C9orf89, MAGI2, MLPH, or RHBDD2 in fibroblasts surrounding carcinoma cells ($n = 60$). The x axis signifies elapsed months from the date of diagnosis. **c** Association between RHBDD2 and CXCL12 expression in fibroblasts surrounding carcinoma cells. The inverse correlation between RHBDD2 and CXCL12 was found (right table). Left column: a representative case of RHBDD2 ( + )/CXCL12 ( − ); right column: a representative case of RHBDD2 ( − ) /CXCL12 ( + ). **d** Schematic representation of RHBDD2–CXCL12 axis in drug resistance with cell–cell interactions. Statistical significance values were calculated by performing log-rank tests (**b**) and Fisher's exact test (**c**). *$P < 0.05$; ***$P < 0.001$; and NS: non-significant. Red arrows indicate fibroblasts. Scale bar, 25 µm.

Library A (Addgene, MA, USA, #1000000049) in lentiGuide-Puro[36] was transduced into Cas9-Dendra2-expressing HEK293T cells and then treated with puromycin (1 µg/ml, InvivoGen) for 2 weeks. The multiplicity of infection (MOI) was calculated and provided at 0.5. A total of $7.14 \times 10^6$ cells of HEK293T transduced with gRNAs were used for screening in $15 \times 96$-well plates (for a total of 1428 wells) per screening. Library transduction and screening were performed in independent two batches. As a control, 12 wells with HEK293T cells without library induction were also conducted in the following experiments. Incubation HEK293T for 48 h to adhere to the bottoms of the plates, then $5 \times 10^3$ cells of U937 were seeded per well, and an additional 48 h later, both U937 and the Dendra2 library-induced HEK293T cells were exposed to 3 µM of cytarabine for 120 h. Regarding the wells where surviving U937 colonies were observed, both surviving U937 and Dendra2 library-induced HEK293T cells were transferred to 24-well plates to expand the scale of Dendra2 library-induced HEK293T, and co-culture was restarted with fresh U937 with cytarabine and then re-scaled up to six-well plates. All supporting cells close to the viable U937 colonies under cytarabine exposure observed in six-well plates were applied to photoconversion. Photo-conversions were performed using FV1200 biological confocal laser scanning microscopes (Olympus, Japan). The supporting cells close to the mulberry-like viable U937 colonies were identified by differential interference images generated by the FV1200. Observations and image captures of the green form of Dendra2 were generated using the 473 nm laser, whereas red form was generated with the 559 nm laser. For photoconversion, the 405 nm laser at maximum power (100%) was used. The duration of the illumination period was set to 60 s, enough to observe the red form of Dendra2 under the 559 nm laser. The dotted lines showed the target area to be illuminated on the operating screen (Fig. 1b, Step 4). Illumination was performed using the "bleach" mode of the FV1200[37]. The whole process of observation and photoconversion was performed manually.

Dendra2 photoconverted HEK293T cells were sorted using the BD FACSAria III (Becton Dickinson and Company, NJ, USA) in a sterile environment. The cells were initially gated based on FSC-A and SSC-A channels to exclude the debris and dead cells. Subsequently, the PI channel-positive cells were sorted as the populations that exhibited negligible signals in the un-photoconverted negative controls. Approximately 1000 photoconverted cells were sorted per each well of the cells in the wells that underwent photoconversion.

The integrated gRNAs were sequenced by the following primers using the Expand High Fidelity PCR system (Sigma-Aldrich): GeCKO 1717F 5′-gagggcctatttcccatgat-3′ and GeCKO 3913 R 5′-cggtgccacttttttcaagtt-3′. Genomic DNA isolations from each well were acquired using a DNeasy Blood & Tissue Kit (QIAGEN, Germany). The fragments were sequenced by TA cloning with pGEM-T Vector Systems (Promega, WI, USA). The analyzed sequences of the gRNA region were compared with a list of human library A gRNA sequences (https://media.addgene.org/cms/filer_public/a4/b8/a4b8d181-c489-4dd7-823a-fe267fd7b277/human_geckov2_library_a_09mar2015.csv) to identify candidate genes.

**Cell viability assessment**. Cell viability under drug exposure was assessed using the MTS assay (Promega) following the manufacturer's protocol.

In the analysis of cell sensitivities to cytarabine or gemcitabine, $1 \times 10^3$ cells were seeded in a 96-well plate for 24 h and then treated with cytarabine or gemcitabine with serial dilution methods. After 48 h of drug exposure, cell viability was evaluated with the MTS assay, and IC50 was estimated.

In the analysis of U937 cell viability cultured with the supernatant from HEK293T, $1 \times 10^6$ cells of HEK293T RHBDD2-CKO were seeded in 2 ml of RPMI in a six-well plate. After 48 h of incubation, the cultured supernatant was collected and filtered. Then, $3 \times 10^4$ cells of U937 were seeded in 150 µl of "collected supernatant of HEK293T RHBDD2-CKO" in a 96-well plate and incubated for 48 h, then treated with 250 nM of cytarabine. After 48 h of cytarabine exposure, cell viability was evaluated using the MTS assay.

**Assessment of Propidium iodide and Annexin V in GFP-positive U937 cells**. In the analysis of Propidium iodide (PI) or Annexin V in GFP-positive U937 cells with cytarabine exposure, $1 \times 10^5$ cells of parental HEK293T cells were seeded in a 24-well plate with the RPMI medium. For 48 h of incubation, $1 \times 10^5$ cells of GFP-positive U937 were added per well. After 48 h of co-culture, cells were exposed to 2.5, 5, or 10 µM of cytarabine for 48 h. As a control, co-culture without cytarabine (cytarabine 0 µM) groups were also conducted in the experiment. After cytarabine

exposure, all cells in each well were collected, then treated with PI (Sigma-Aldrich) or Annexin V-PE (Medical & Biological Laboratories, Japan) following the manufacturer's protocol. The percentages of PI-positive cells or Annexin V-positive cells were evaluated using the BD FACSCanto II (Becton Dickinson and Company). The GFP-positive U937 cells were gated and evaluated expressions of PI or Annexin V.

**Cell co-culture validation experiments**. The sequences of all oligonucleotides used in the experiments are provided in Supplementary Table 1 and Supplementary Data 3. The designed oligonucleotides were purchased from Invitrogen. Each oligonucleotide of all candidate genes and non-targeted gRNA (Non-target) were integrated into lentiCRISPRv2 plasmids following the manufacturer's protocol.

HEK293T or UE7T-9 were transduced with lentiCRISPRv2 of each candidate gene In subsequent experiments, HEK293T or UE7T-9 cells transduced with lentiCRISPRv2 Non-target were used as controls. After a 2-week selection with puromycin, cells were conducted in the cell co-culture validation experiments. For labeling for the indication of live tumor cells, plentiPGK-GFP-pgk-Hygro was induced to U937, and plentiCMV-GFP-Puro was induced to MIA PaCa-2 and SUIT-2. LentiU6-DCK-Hygro was generated with lentiGuide-puro by replacing the sequence of the puromycin-resistant gene with the hygromycin-resistant gene and the integrated target gRNA sequence for DCK (Supplementary Data 3 and previous report[38]). LentiU6-DCK-Hygro was transduced to Cas9-HEK293T. The HEK293T DCK-CKO clone was established with limiting dilution methods and then transduced with lentiCRISPRv2-gRNA to make knockout HEK293T cells of each candidate gene.

In the co-culture experiments of mutant-HEK293T with U937 or mutant-UE7T-9 with U937 cells, $1 \times 10^5$ cells of the target mutated HEK293T or target mutated UE7T-9 cells were seeded in a 24-well plate with the RPMI medium. After 48 h of incubation, $1 \times 10^5$ cells of GFP-positive U937 were added per well. After an additional 48 h of co-culture to make sure all cells could contact each other, cells were exposed to 5 µM of cytarabine for 48 h.

In the co-culture experiments of mutant-HEK293T with pancreatic cancer cells, $1 \times 10^5$ cells of DCK and target mutated HEK293T cells and $1 \times 10^5$ cells of GFP-positive pancreatic cancer cells (MIA PaCa-2 or SUIT-2) were seeded in 24-well plates and co-cultured (with the EMEM medium for MIA PaCa-2 or the RPMI medium for SUIT-2). After 48 h of co-culture, cells were exposed to gemcitabine (MIA PaCa-2: 10 µM, SUIT-2: 3 µM) for 168 h.

For each co-culture experiment, three wells were prepared with the same conditions, and three fields were randomly captured for each well with the drug exposure. The observers were blinded to sample identity and selected observation fields of view randomly. Fluorescent image capture was performed using an inverted fluorescence phase-contrast microscope BZ-X810 (KEYENCE, Japan). The field of view was ×20 objective using a green laser. The count of viable cells per field of view was performed by different blinded observers from observers capturing images using Image J (https://imagej.net/software/fiji/)[39]. To compare viability, the number of viable cells relative to that co-cultured with the control (Non-target) of supporting cells was calculated.

**Confirmation of knockouts of candidate genes in CKO clones**. To confirm the knockouts of candidate genes in HEK293T-CKO clones or UE7T-9 CKO clones, western blot analysis or flow cytometric analysis were performed. Western blot analysis was applied for HEK293T cells of DCK-CKO, C9orf89-CKO, C19orf70-CKO, C21orf33-CKO, and RHBDD2-CKO, as well as UE7T-9 cells of C9orf89-CKO, MLPH-CKO, and RHBDD2-CKO. In the western blot analysis, cells were lysed and sonicated in the SDS-PAGE sample buffer. Protein samples were separated by SDS-PAGE and then transferred to PVDF membranes. β-actin was used as the loading control. Membranes were incubated overnight with primary antibodies (anti-C9orf89, 1:1000; anti-C19orf70, 1:1000; anti-C21orf33, 1:1000; anti-DCK, 1:1000; anti-MLPH, 1:1000; and anti-RHBDD2, 1:1000) at 4 °C. Anti-rabbit or mouse HRP-linked IgG antibody was used as a secondary antibody (1:5000, Cytiva, MA, USA). Detection was performed using Clarity Western ECL Substrate (Bio-Rad Laboratories, CA, USA). The blot was imaged using the ChemiDoc MP Imaging System with Image Lab software ver. 4.1 (Bio-Rad Laboratories).

Flow cytometric analysis was applied to HEK293T cells of MAGI2-CKO and MLPH-CKO, as well as the UE7T-9 cell of MAGI2-CKO owing to their difficulty of detection with western blot in the parental cells. Cells were fixed with 4%

paraformaldehyde, and permeabilized with 90% methanol following the manufacturer's protocol. All collected cells were incubated for 1 h with anti-MAGI2 (1:200) or anti-MLPH (1:200) primary antibodies at room temperature, washed twice with PBS, and incubated with anti-rabbit IgG (H + L), F(ab')2 Fragment (PE conjugate) (Cell Signaling Technology, MA, USA) for 30 min at room temperature and protected from light. Then, they were washed twice with PBS. The expression of MAGI2 or MLPH were evaluated using the BD FACSCanto II. The cells were gated based on FSC-A and SSC-A to exclude debris and dead cells. Subsequently, cells were evaluated with PE signals by comparing them with stained control cells and unstained negative controls.

Information about all primary antibodies used in all experiments is provided in Supplementary Table 4.

**The analysis of cleaved caspase-3 in the co-culture system.** In the analysis of cleaved caspase-3 in the HEK293T-U937 co-culture system, $5 \times 10^5$ HEK293T-CKO cells were seeded in six-well plates. After 48 h of incubation, $5 \times 10^5$ cells of U937 were added per well. After an additional 48 h of co-culture, cells were exposed to 5 μM of cytarabine for 48 h. As a control, HEK293T mono-culture groups were also conducted in the experiment. After 48 h of cytarabine exposure, all the cells in each well were collected, fixed with 4% paraformaldehyde, and permeabilized with 90% methanol following the manufacturer's protocol. All collected cells were incubated for 1 h with anti-cleaved caspase-3 primary antibodies (1:6400) at room temperature, washed twice with PBS, and incubated with anti-rabbit IgG (H + L), F(ab')2 Fragment (PE conjugate) for 30 min at room temperature and protected from light. Then, they were washed twice with PBS. The percentages of cleaved caspase-3-positive cells were evaluated using the BD FACSCanto II. The cells were gated based on PE-A and SSC-A channels as populations that exhibited negligible signals in the unstained negative controls.

**Assay of secreted CXCL12 levels in CKO clones.** To confirm the upregulation of secretion of CXCL12 in CKO clones of HEK293T and UE7T-9 cells, an enzyme-linked immunosorbent assay (ELISA) was conducted. HEK293T or UE7T-9 cells of *C9orf89*-CKO, *MAGI2*-CKO, *MLPH*-CKO, and *RHBDD2*-CKO, and Non-target were seeded in 350 μL of DMEM in a 24-well plate (HEK293T: $4 \times 10^5$ cells/well, UE7T-9: $2 \times 10^5$ cells/well). After 48 h of incubation, each cell culture medium was collected and analyzed. The secreted CXCL12 level was assessed using a Human CXCL12/SDF-1 Alpha Quantikine ELISA Kit (R&D Systems), following the manufacturer's protocol.

**The analysis of cleaved caspase-3 with the neutralization of CXCL12.** In the experiments on the neutralization of CXCL12, $1 \times 10^5$ cells of HEK293T/UE7T-9 *RHBDD2*-CKO or Non-target were seeded in 350 μl of fresh RPMI medium in a 24-well plate. After 48 h of incubation, $1 \times 10^5$ cells of U937 were added in 350 μl of fresh RPMI medium with human/mouse CXCL12/SDF-1 antibody or mouse IgG1 isotype control (25 μg/ml, R&D Systems, UK). After 48 h of co-culture, cells were exposed to 5 μM of cytarabine for 48 h. After 48 h of cytarabine exposure, all the cells in each well were collected, fixed with 4% paraformaldehyde, and permeabilized with 90% methanol following the manufacturer's protocol. All collected cells were incubated for 1 h with anti-cleaved caspase-3 primary antibodies (1:6400) at room temperature, washed twice with PBS, and incubated with anti-rabbit IgG (H + L), F(ab')2 Fragment (PE conjugate) for 30 min at room temperature and protected from light. Then, they were washed twice with PBS. The percentages of cleaved caspase-3-positive cells were evaluated using the BD FACSCanto II. The cells were gated based on PE-A and SSC-A channels as populations that exhibited negligible signals in the unstained negative controls.

**The analysis of phospho-Akt.** In the analysis of phospho-Akt of U937 co-cultured with HEK293T or UE7T-9, U937 was labeled with GFP to evaluate phospho-Akt in U937 cells only. Then, $5 \times 10^5$ cells of HEK293T/UE7T-9 *RHBDD2*-CKO or Non-target cells were seeded in 2 ml of RPMI in a six-well plate. After 48 h of incubation, all mediums were changed with $5 \times 10^5$ cells of GFP-positive U937 in 2 ml of RPMI medium. As a control, U937 mono-culture groups were also conducted in the experiment. After 48 h of co-culture, all cells in each well were collected, fixed with 4% paraformaldehyde, and permeabilized with 0.1% of Triton X-100 (Sigma-Aldrich) following the manufacturer's protocol. All collected cells were incubated for 1 h with anti-phospho-Akt primary antibodies (1:100) at room temperature, washed twice with PBS, and incubated with anti-rabbit IgG (H + L), F(ab')2 Fragment (PE conjugate) for 30 min at room temperature and protected from light. Then, they were washed twice with PBS. The expression of phospho-Akt in U937-GFP-positive cells was evaluated using the BD FACSCanto II. The cells were gated based on FSC-A and SSC-A channels to exclude debris and dead cells, and they were further gated based on the GFP-A channel to exclude HEK293T or UE7T-9. The mean fluorescence of PE in the GFP-positive population was evaluated as the expression of phospho-Akt in U937-GFP-positive cells. Gating strategy is provided in Supplementary Fig. 15.

**RNA sequencing and analysis.** Total RNA was extracted from HEK293T or UE7T-9 cells of *C9orf89*-CKO, *MAGI2*-CKO, *MLPH*-CKO, *RHBDD2*-CKO, and Non-target incubated in three wells of six-well plates using the RNeasy Mini Kit (QIAGEN) following the manufacturer's protocol, including the DNase I

treatment. All RNAs were analyzed by the RNA sequencing service provided by Novogene Co., Ltd. (Beijing, China) using the NovaSeq 6000 Sequencing System (Illumina, San Diego, CA, USA) and 6 Gb of the Eukaryotic mRNA Library. The RNA profile data were deposited in a public database, Gene Expression Omnibus (GEO), under access number GSE 203256. For GSEA, normalized expression data were analyzed and visualized with GSEA software (version 4.2.0, https://www.gsea-msigdb.org/gsea/index.jsp). The normalized enrichment score (NES), nominal *P*-value (NOM-*P*), and false discovery rate *q*-value (FDR-*q*) were calculated for comparison, and the categories selected were universally upregulated in HEK293T cells of *C9orf89*-CKO, *MAGI2*-CKO, *MLPH*-CKO, and *RHBDD2*-CKO. The relative enrichment of individual genes was assessed based on the rank metric score following the GSEA. Volcano plots were generated using Novosmart software (Novogene).

In the assessment of upregulation of *CXCL12* in HEK293T-CKO and UE7T-9-CKO clones, a quantitative polymerase chain reaction (qPCR) was performed based on SYBR Green (Toyobo, Japan), and the expression of *CXCL12* was normalized to β-actin levels. Primers for *CXCL12* or β-actin are shown in the following—CXCL12 forward: ATGAACGCCAAGGTCGTG, CXCL12 reverse: ACATGGCTTTCGAAGAATCG, β-actin forward: CACAGAGCCTCGCCTTTGCC, and β-actin reverse: ACATGCCGGAGCCGTTGTC. All assays were performed on an ABI 7900 system and analyzed with ABI 7900 SDS software v.2.4.1 (Thermo Fisher Scientific).

**Analysis of cleaved caspase-3 with recombinant CXCL12.** In the experiments with recombinant CXCL12, $1 \times 10^5$ cells of parental HEK293T or UE7T-9 cells were seeded in a 24-well plate. After 48 h of incubation, $1 \times 10^5$ cells of U937 were added to 1000 μl of fresh RPMI medium with or without recombinant Human/Rhesus Macaque/Feline CXCL12/SDF-1 alpha (1000, 10,000, 100,000 pg/μL, R&D Systems). U937 mono-culture groups with recombinant CXCL12 were also conducted in the experiment. After 48 h of co-culture or mono-culture, cells were exposed to 5 μM of cytarabine for 48 h. After 48 h of cytarabine exposure, all the cells in each well were collected, fixed with 4% paraformaldehyde, and permeabilized with 90% methanol following the manufacturer's protocol. All collected cells were incubated for 1 h with anti-cleaved caspase-3 primary antibodies (1:6400) at room temperature, washed twice with PBS, and incubated with anti-rabbit IgG (H + L), F(ab')2 Fragment (PE conjugate) for 30 min at room temperature and protected from light. Then, they were washed with PBS. The percentages of cleaved caspase-3-positive cells were evaluated using the BD FACSCanto II. The cells were gated based on PE-A and SSC-A channels as populations that exhibited negligible signals in the unstained negative controls.

**Immunohistostaining of clinical samples.** Sixty clinical samples of invasive pancreatic ductal carcinoma were obtained from 60 patients who underwent operations without neoadjuvant chemotherapy at the Tokyo Medical and Dental University Hospital, Tokyo, between 2008 and 2016. Specimens were obtained by surgical resection, fixed in 10% neutralized formalin, and embedded in paraffin according to routine protocols used in conventional histopathological examinations. Informed consent was obtained from all patients by an opt-out method, and the study was approved by the ethics committees of Tokyo Medical and Dental University. All procedures were performed following the ethical standards established by these committees (M2000-1458-07).

Formalin-fixed, paraffin-embedded (FFPE) tissues were sectioned at a thickness of 4 μm, placed on silane-coated slides, and deparaffinized. Heat-based antigen retrieval, endogenous peroxidase blockade using 3% hydrogen peroxide, and blocking were performed with normal sera. The sections were incubated overnight with primary antibodies against C9orf89 (1:200), CXCL12 (1:100), MAGI2 (1:200), MLPH (1:800), and RHBDD2 (1:200) at 4 °C. Primary antibodies were detected using an ABC kit (Vector Laboratories, CA, USA). Color development was performed using diaminobenzidine (Vector Laboratories). The scoring of immunostainability was confirmed by two pathologists (K. Sugita and M. Kurata) based on the following criteria: not stained in fibroblasts was (−), positive in only a few fibroblasts or weakly positive were (1 +), and strongly positive on most of the fibroblasts was (2 +). Representative images of each antibody are shown. Based on scoring, (−) and (1 +) were classified as negative groups, and (2 +) was the positive group (Supplementary Fig. 16).

The following five parameters were evaluated as clinicopathologic factors: sex (female vs. male), age (≤ 70 y vs. > 70 y), pT category (1–2 vs. 3–4), tumor size (≤20 mm vs. >20 mm), and UICC stage (I–IIB vs. III–IV).

**Statistics and reproducibility.** One-way analysis of variance (ANOVA) with Dunnett's test or Bonferroni's test was used for the analysis of data involving multiple groups. The two-tailed unpaired Student's *t* test was used to compare the two groups. OS was determined from the date of diagnosis and that of last follow-up or death. Kaplan–Meier survival curves were used to estimate rates of OS. The log-rank test was used to analyze differences in survival between the groups. Univariate and multivariate analyses were performed using the Cox proportional hazard regression model. Correlations between the two groups were assessed using Fisher's exact test. All *P* values < 0.05 were considered statistically significant. In GSEA, results were considered significant if both NOM-*P* < 0.05 and FDR-*q* < 0.25 were satisfied. In all graphs, the data are presented as mean ± standard deviation

(SD). Meanwhile, the results were obtained independently in triplicate or quin-tuplicate for each experiment, and the experiments were usually repeated two or three times. All statistical analyses were performed using the free statistical software EZR (version 1.40)[40].

**Reporting summary**. Further information on research design is available in the Nature Portfolio Reporting Summary linked to this article.

## Data availability
Source data used in Figs. 2–4 are provided in Supplementary Data 4. Source data of Fig. 5b are provided in Supplementary Data 2. All other data generated, including raw data supporting, in this study have been deposited in figshare.com. Figure 1 (https://figshare.com/s/3092ec8b6e9698f7d2a7), Supplementary Fig. 5 (https://figshare.com/s/ec9e74e885b91d41dcef), Supplementary Figs. 8, 9, 11, 12, (https://figshare.com/s/ce402d008054d7d9a333), plasmid maps (https://figshare.com/s/b9c98a0206a6972ea0ed), and all other revised raw data (10.6084/m9.figshare.22492372). RNA-seq data have been deposited in Gene Expression Omnibus (GEO) under access number GSE 203256. Following plasmids have been deposited; pDendra2-Hygro (Addgene ID:#202407) and pLenti-DCK-Hygro (Addgene ID:#202409).

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

## Acknowledgements
The authors would like to thank Soichiro Yoshida from the Department of Urology, Graduate School of Medical and Dental Sciences, Tokyo Medical and Dental University, and the members of the stem cell laboratory of Tokyo Medical and Dental University for their technical assistance and advice. Funding for this project was provided by Grant-in-Aid for Scientific Research (C) (21K06945) from Japan Society for the Promotion of Science.

## Author contributions
M. Kurata conceived this project. K. Sugita and M. Kurata designed the study, performed and coordinated the experiments, and wrote the manuscript. K. Sugita, R. Nakayama, S.I., M. Ikeda, R. Narita, S.O., K. Shimizu, S. Saito, and B.S.M. performed wet-lab experiments. K. Sugita, S.O., M. Inoue, and M. Kurata analyzed clinical samples. K. Sugita, R. Nakayama, R. Narita, S.O., K. Shimizu, and M. Kurata performed statistical analysis. I.O., S. Sato, K.Y., B.S.M., D.A.L., and M. Kitagawa contributed to project coordination and provided overall supervision. All authors reviewed and approved the final manuscript.

## Competing interests
D.A.L. is the co-founder and co-owner of NeoClone Biotechnologies, Inc., Discovery Genomics, Inc. (acquired by Immunsoft, Inc.), B-MoGen Biotechnologies, Inc. (acquired by Bio-Techne corporation), and Luminary Therapeutics, Inc. D.A.L. holds equity in, is a Board of Directors member of, and serves as the Senior Scientific Advisor to Recombinetics, a genome-editing company, and Makana, a xenotransplantation company. The business of all the companies above is unrelated to the contents of this manuscript. No disclosures were reported by the other authors.

## Additional information

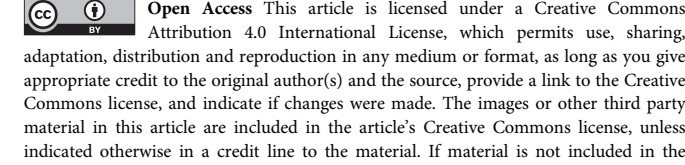

