## [Peer Review File · Communications Biology]

Reviewers' comments:

Reviewer #1 (Remarks to the Author):

This study introduces an interesting approach to investigate drug resistance, caused by supporting cells of the tumor microenvironment that are found in proximity/physical interaction with tumor cells. In their method the authors transduced HEK293 cells with a whole genome library of gRNAs. The HEK293 cells were cocultured with a leukemia cell line in the presence of a drug in small wells, which were then selected for viable tumor cells. Cells were collected and plated for expansion. After expansion, the HEK293 cells in the same location with drug-resistant (live) tumor cells were marked, using the Dendra2 protein that changes color from green to red, once exposed to UV light. The HEK293 cells were then sorted for red fluorescence. The authors identified a set of 39 candidates, and generated single knockouts per candidate, testing them in coculture with tumor cells. 11 of these single knockouts generated higher number of viable cells compared with non-targeting knockouts in both HEK293 and UE7T. These 11 knockouts were further tested in coculture with pancreatic cell lines treated with a drug and tested for viability compared with non-targeting knockouts and for cleaved caspase 3 activity in HEK293-U937 cocultures. These results narrowed down the list of candidate knockouts to 4. HEK293 with these knockouts were then profiled by RNAseq to identify possible mechanism for resistance which suggested CXCL12 as a possible regulator of drug resistance through the PI3K-AKT-mTOR pathway.

The method presented by the authors is unique and novel, however, the CRISPR data provided by the authors is lacking and the data supporting the suggested mechanism is not strong enough.

1. The coverage of gRNA to cells is on the lower side of ~100 fold.
2. There are no files in the results for the sequencing of the different gRNAs, and there are no graphs that show the distribution of the guides before and after coculture. It is unclear how many guides showed depletion, enrichment after treatment. Also, there is no comparison between the two replicates and how much overlap there is between them.
3. How many of the 39 genes have no expression HEK cells?
4. The authors mention that they sorted 1000 genes from each 6 well, it is unclear why they chose such low number of cells, considering the huge loss of cells during sorting.
5. The authors suggest that drug resistance happens through cytokine production of CXCL12 but the supporting data for that is not convincing enough. What are the CXCL12 levels secreted by the different knockouts in HEK193, UE7T? Does coculture of HEK293, UE7T with CXCL12 recombinant protein generates the same observed drug resistance of the tumor cells? can CXCL12 on its own generate drug resistance in tumor cells? How do the authors explain the absence of CXCL12 from the list of knockout candidates? Would CXCL12 knockout in HEK293, UE7T generate the same results?
6. Do the different knockouts in HEK293 also show resistance to higher doses of the drug in monocultures without tumors?
7. What do the x-axis in figure 4f represent? Please provide the scatter plots by flow. What is the gating strategy?
8. Please provide a tab delimited summary file for the gene level RNAseq data
9. Please provide cell numbers for figures 2b,c, 3

Reviewer #2 (Remarks to the Author):

The manuscript by Sugita and coauthors reports an indirect CRISPR screening system with

photoconvertible fluorescent protein Dendra2 to screen for key factors responsible for drug resistance with cell-cell interactions. The authors first applied this indirect CRISPR screening approach in HEK293T-U937 co-culture system and found 39 candidate genes with drug resistance functions in supporting cells. Then they validated these candidate genes in several different co-culture systems, investigated their biological features, and found a new axis linking RHBDD2, CXCL12, and PI3k-Akt-mTOR which is responsible for anticancer drug resistance. Finally, they analyzed the expression level of four important candidate genes in clinical pancreatic cancer samples to show their significance as prognostic factors. Overall, this manuscript is well-written with clear rationale and detailed methods for reproducing the work. The indirect CRISPR screening system is novel with good potential to be applied for screening in other cell-cell interaction systems. The discovery of RHBDD2-CXCL12 axis is also new and interesting. The reviewer only has several concerns.

Concerns:

1. In Fig. 2 and Supplementary Figure 3, the viability of U937 cells in the co-culture system was assessed by quantifying GFP-positive cells as viable U937 cells. The reviewer is concerned if the GFP-positive cells can correctly represent viable cells.
2. There are 39 candidate genes identified by screening. However, only 11 genes were successfully validated with drug resistance functions (Fig. 2). How do you explain why the other 28 candidate genes were identified by screening but not successfully validated? Does that mean the indirect CRISPR screening assay has a high false positive ratio?
3. There is no clear description and citation of the CRISPR knockout library used for the indirect CRISPR screening. Please add this information in the manuscript.
4. For line 120-121, please describe how the conditions for drug selection was tightened.
5. Please include a calibration bar in Supplementary Figure 1 to show the relative intensity of the red channel.
6. It's better to include a diagram to describe the newly discovered axis responsible for anticancer drug resistance in the main figure to help the readers to understand.
7. For line 103, please include a comma after "As we expected".

Manuscript ID: COMMSBIO-22-3524-T

Indirect CRISPR screening with photoconversion revealed key factors of drug resistance with cell–cell interactions

We would like to thank the reviewers for their thoughtful and thorough analysis of our manuscript. We have endeavored to address all of the issues they presented. The response comments are in **bold and blue**.

Reviewer #1 (Remarks to the Author):

This study introduces an interesting approach to investigate drug resistance, caused by supporting cells of the tumor microenvironment that are found in proximity/physical interaction with tumor cells.

In their method the authors transduced HEK293 cells with a whole genome library of gRNAs. The HEK293 cells were cocultured with a leukemia cell line in the presence of a drug in small wells, which were then selected for viable tumor cells. Cells were collected and plated for expansion. After expansion, the HEK293 cells in the same location with drug-resistant (live) tumor cells were marked, using the Dendra2 protein that changes color from green to red, once exposed to UV light. The HEK293 cells were then sorted for red fluorescence. The authors identified a set of 39 candidates, and generated single knockouts per candidate, testing them in coculture with tumor cells. 11 of these single knockouts generated higher number of viable cells compared with non-targeting knockouts in both HEK293 and UE7T. These 11 knockouts were further tested in coculture with pancreatic cell lines treated with a drug and tested for viability compared with non-targeting knockouts and for cleaved caspase 3 activity in HEK293-U937 cocultures. These results narrowed down the list of candidate knockouts to 4 HEK293 with these knockouts were then profiled by RNAseq to identify possible mechanism for resistance which suggested CXCL12 as a possible regulator of drug resistance through the PI3K-AKT-mTOR pathway.

The method presented by the authors is unique and novel, however, the CRISPR data provided by the authors is lacking and the data supporting the suggested mechanism is not strong enough.

Thank you for your thoughtful analysis of our manuscript and appreciation of the novelty of our study, noting that although there are published works on various CRISPR screening

methods, this method is *unique and novel*. Thank you also generally for your time and valuable feedback.

1. The coverage of gRNA to cells is on the lower side of ~100 fold.

Response: Thank you for pointing out. Human CRISPR Knockout Pooled Library A contains 65,384 (6.54×10^4) gRNAs, and the number of viral-infected HEK293T cells at 0.5 MOI applied in each co-culture experiment is 7.14×10^6 cells. The number of viral-infected cells used in screening theoretically exceeds by 100-fold the number of gRNAs; thus, the diversity of the CRISPR library might be preserved.

The revised part is highlighted in the manuscript as follows:

Methods (Line 390-394)

The multiplicity of infection (MOI) was calculated and provided at 0.5. A total of 7.14×10^6 cells of HEK293T transduced with gRNAs were used for screening in 15×96 -well plates (for a total of 1428 wells) per screening. Library transduction and screening were performed in independent two batches.

2. There are no files in the results for the sequencing of the different gRNAs, and there are no graphs that show the distribution of the guides before and after coculture. It is unclear how many guides showed depletion, enrichment after treatment. Also, there is no comparison between the two replicates and how much overlap there is between them.

Response: Thank you for the important insights. The screening methods developed in the present study were based on microscope observation and manual photoconversion, and we recognized in the pre-screening that there was a numerical limit to the number of cells that could be labeled and collected. Therefore, we did not analyze gRNAs of the co-cultured cells using NGS and, unfortunately, we did not store those samples suitably for DNA extraction of co-cultured cells as part of the present study. Therefore, we could not perform sequencing of the different gRNAs before and after coculturing.

There are zero overlaps in the identified genes between the two independent replicates (Table 1); this may be because the number of candidate genes is small in the present study.

The present study aimed to develop indirect screening methods using photoconversion to clarify indirect cell–cell interaction mechanisms. Fortunately, positive candidates were obtained from the screening. We validated their ability to induce drug resistance and analyzed their biological function. We recognize that the most critical limitation of this research is the numerical limitation caused by microscopic observation and manual illumination with labeling and separating of target cells. We would like to develop a high-throughput screening system that automatically recognizes the target cells, automatically illuminates them with a laser, and

separates all of them, thereby enabling us to process a large number of cells without losing any candidates.

We are planning to develop such a high-throughput screening system, confirm the diversity of all cells during screening, collect all labeled cells, and analyze those collected cells using NGS. By applying such improved methods, we hope to elucidate mechanisms of cell–cell interactions more efficiently.

3. How many of the 39 genes have no expression HEK cells?

Response: Gene expression in HEK293T was defined based on transcriptional product expression data listed in the cell line RNA database of The Human Protein Atlas (<https://www.proteinatlas.org/>). A total of 27 genes have expression, transcriptional production (nTPM) ≥ 0.1 , in HEK293T cells, and 12 genes have no expression in HEK293T cells. All values of nTPM are shown in the following table (for reviewers and editors).

At the early stage of the present study, when 39 candidate genes were obtained from indirect CRISPR screening, even if the expression of candidate genes was low in HEK293T, we were interested in the repeatability of the indirect CRISPR screening itself and also considered the possibility that they were located upstream of drug-resistance genes and might affect downstream genes. Therefore, regardless of RNA expression levels, all candidate genes were included in validation experiments using multiple types of cells.

At the final selection stage after the validation experiments, candidate genes with strong RNA expression in HEK293T were selected to analyze their biological function. Then, candidate genes with stronger effects were evaluated considering other factors, such as clinical prognosis.

Candidate genes	nTPM	Candidate genes	nTPM
ACTRT2	0	OR4M2	0
AQP6	0.3	OR5J2	0
ATP4A	0	OR5R1	0
C9orf89	12.3	OTUB2	1.5
C10orf118	7.3	OXSM	16
C19orf70	311.9	PDZD8	11.8
C21orf33	77.7	PLAS3	18.3
CHMP6	23.8	RAC2	0.6
ENAM	0	RHBDD2	46.3
ERVV-1	0	S100A4	2.3
ESYT1	54.2	SCHIP1	27.6
FAM25A	0	SETD9	6.9
IL2RG	0.2	SLCO1B1	0
KCNAB1	1	STAP1	0
KCNJ5	0	TMEM249	0.1
KRT14	0	VIL1	0.4
LRRC40	20	ZFAND6	107.2
MAGI2	0.8	ZFP62	19
MLPH	0.3	ZNF48	20.2
ODF3L2	0.1		

4. The authors mention that they sorted 1000 genes from each 6 well, it is unclear why they chose such low number of cells, considering the huge loss of cells during sorting.

Response: Yes, we agree this point. The scenario stands as a technical limitation of the present study. In the present screening system, there were live U937 colonies that were observed under the microscope to be “mulberry-like” under cytarabine exposure.

During the present drug selection, mulberry-like U937 cell colonies were identified and the drug-resistance-inducible HEK293T cells also proliferated continuously at the bottom of the colonies. In that scenario, we deemed that collecting all drug-resistance-inducible cells was not necessary, with it instead appropriate to sample just a portion of drug-resistance-inducible HEK293T cells in the present study with technical limitations. The 1000 sorted cells might provide coverage for the drug-resistance-inducible -phenotype cells we sought to collect.

We manually photoconverted as many of these viable colonies as possible using laser illumination. We will sort the photoconverted red form of Dendra2 more efficiently with automatic illuminating systems, meaning it will be possible to sort a greater amount in the future.

5. The authors suggest that drug resistance happens through cytokine production of CXCL12 but the supporting data for that is not convincing enough. What are the CXCL12 levels secreted by the different knockouts in HEK293, UE7T? Does coculture of HEK293, UE7T with CXCL12 recombinant protein generate the same observed drug resistance of the tumor cells? Can CXCL12 on its own generate drug resistance in tumor cells? How do the authors explain the absence of CXCL12 from the list of knockout candidates? Would CXCL12 knockout in HEK293, UE7T generate the same results?

Response: Thank you for your thoughtful consideration. We added new data on secretion levels of CXCL12 in HEK293T and UE7T-9 CKO clones (Figure 4f, g) obtained by using an enzyme-linked immunosorbent assay (ELISA). CXCL12 secretion was up-regulated universally in *C9orf89*-CKO, *MAGI2*-CKO, *MLPH*-CKO, and *RHBDD2*-CKO clones of HEK293T and UE7T-9 cells.

Figure 4f, g

Figure 4 f, g

CXCL12 secretion was significantly up-regulated in HEK293T (f) and UE7T-9 (g) cells of *C9orf89*-CKO, *MAGI2*-CKO, *MLPH*-CKO, and *RHBDD2*-CKO.

We also added new data for the co-culture experiments on the analysis of cleaved caspase-3 under cytarabine (Figure 4i) and the analysis of phospho-Akt (Figure 4k), using *RHBDD2*-CKO UE7T-9 instead of *RHBDD2*-CKO HEK293T. In the analysis of cleaved caspase-3 under cytarabine, significant inhibition of apoptosis was observed in the group co-cultured with *RHBDD2*-CKO UE7T-9 compared to the control. Furthermore, this inhibition of apoptosis observed in the co-culture experiment with *RHBDD2*-CKO UE7T-9 was

canceled by treatment with CXCL12-neutralizing antibodies. In the analysis of phospho-Akt, phospho-Akt was increased in U937 co-cultured with *RHBDD2*-CKO UE7T-9. These results of co-culture experiments with *RHBDD2*-CKO UE7T-9 were similar to those observed when U937 cells were co-cultured with *RHBDD2*-CKO HEK293T.

Figure 4i, k

Figure 4 i, k
 h, i In the co-culture experiment of U937 with *RHBDD2*-CKO HEK293T (h) or *RHBDD2*-CKO UE7T-9 (i), pre-treatment with CXCL12-neutralizing antibodies increased cleaved caspase-3-positive cells under cytarabine exposure for 48 h, which was similar to the effect observed when the cells were co-cultured with control HEK293T or UE7T-9. j, k Evaluation of phospho-Akt downstream of CXCL12 in GFP-positive U937 under mono- or co-culture conditions. Phospho-Akt was increased in U937 co-cultured with *RHBDD2*-CKO HEK293T (j) and *RHBDD2*-CKO UE7T-9 (k) for 48 h.

Additionally, the baseline of secreted CXCL12 in UE7T-9 cells was approximately 1000 pg/mL (Figure 4g), and that background secretion of CXCL12 may have affected the co-culture system. In the U937-UE7T-9 Non-target co-culture experiments for cleaved caspase-3 under cytarabine, cleaved caspase-3-positive cells under cytarabine exposure were increased in the treatment group with CXCL12-neutralizing antibodies compared to the group with the isotype control (Figure 4i). In the analysis of phospho-Akt, phospho-Akt was increased in U937 co-cultured with the *RHBDD2*-CKO Non-target group compared to the U937 mono-culture group (Figure 4k).

Moreover, we added new data on cytarabine exposure experiments with recombinant CXCL12 (Supplementary Figure 11). In the co-culture experiments of parental

HEK293T-U937 and parental UE7T-9-U937, pre-treatment with recombinant CXCL12 decreased cleaved caspase-3-positive cells under cytarabine exposure (Supplementary Figure 11a, b). In the U937 mono-culture experiment with recombinant CXCL12, pre-treatment with recombinant CXCL12 slightly decreased cleaved caspase-3-positive cells under cytarabine exposure (Supplementary Figure 11c), suggesting that CXCL12 can induce some cytarabine resistance even when up-regulated only in U937 itself.

Supplementary Figure 11.

Supplementary Figure 11. Cytarabine exposure experiment with recombinant CXCL12. (a, b) HEK293T-U937 and UE7T-9-U937 co-culture experiments with recombinant CXCL12. In the co-culture experiments of HEK293T-U937 (a) and UE7T-9-U937 (b), pre-treatment with recombinant CXCL12 decreased cleaved caspase-3-positive cells under cytarabine exposure for 48 h. (c) U937 mono-culture experiment with recombinant CXCL12. Pre-treatment with recombinant CXCL12 decreased cleaved caspase-3-positive cells under cytarabine exposure for 48 h. The experiment was performed in the two independent experiments. Data are represented as mean \pm SD. Statistical significance values were calculated by performing one-way ANOVA with Dunnett's test (a, b) or two-tailed unpaired Student's t-tests (c). ** $p < 0.01$, *** $p < 0.001$

The equivalent concentration of CXCL12 secreted from HEK293T CKO clones (approximately 1000 pg/mL) or UE7T-9 CKO clones (approximately 10,000 pg/mL) less effectively, but significantly, decreased cleaved caspase-3-positive cells under cytarabine exposure (Supplementary Figure 11a, b). This indicated that cell-cell interactions between neighboring cells might be important in inducing anticancer drug resistance in the present system. Candidate genes might induce drug resistance in tumor cells as a result of knockout in peritumoral supporting cells. Since knockout of the four main candidates in our experiments, *C9orf89*, *MAGI2*, *MLPH*, and *RHBDD2*, up-regulated CXCL12 in peritumoral supporting cells, it is appropriate that CXCL12 is not listed among the knockout candidates.

Experiments on the knockout of *CXCL12* in HEK293 and UE7T-9 are interesting; however, we deemed that co-culture experiments with CXCL12-neutralizing antibodies would provide more functional cancellation of CXCL12. Canceling CXCL12 with CXCL12-neutralizing

antibodies results in increases in cleaved caspase-3 in co-culture experiments of U937 with HEK293T *RHBDD2*-CKO, as well as in experiments of U937 with UE7T-9 *RHBDD2*-CKO (Figure 4h, i).

The revised parts are highlighted in the manuscript as follows:

Results

“UE7T-9” was added to Line 233, Line 238, Line 252 , Line 254, Line 256 , Line 259.

(Line 219-221)

Secretion levels of CXCL12 were up-regulated universally in *C9orf89*-CKO, *MAGI2*-CKO, *MLPH*-CKO, and *RHBDD2*-CKO clones of HEK293T and UE7T-9 cells (Fig. 4f, g).

(Line 239-249)

Co-culture experiments of parental HEK293T-U937 cells and parental UE7T-9-U937 cells with cytarabine exposure after pre-treatment with the recombinant CXCL12 were also performed.

The results showed that cytarabine exposure following the treatment with recombinant

CXCL12 decreased cleaved caspase-3-positive cells under cytarabine exposure, which was

similar to the effect observed when the cells were co-cultured with HEK293T *RHBDD2*-CKO

or UE7T-9 *RHBDD2*-CKO (Supplementary Fig. 11a, b). In addition, in the U937 mono-culture

experiment, cleaved caspase-3-positive cells under cytarabine exposure were slightly but

significantly decreased (Supplementary Fig. 11c). These results suggest that a high

concentration of CXCL12 secreted from the supporting cells might induce cytarabine

resistance in U937.

Methods

“UE7T-9” was added to Line 503, Line 513, Line 514, Line 526.

(Line 491-499)

Assay of secreted CXCL12 levels in CKO clones.

To confirm the up-regulation of secretion of CXCL12 in CKO clones of HEK293T and

UE7T-9 cells, an enzyme-linked immunosorbent assay (ELISA) was conducted. HEK293T or

UE7T-9 cells of *C9orf89*-CKO, *MAGI2*-CKO, *MLPH*-CKO, and *RHBDD2*-CKO, and

Non-target were seeded in 350 μ L of DMEM in a 24-well plate (HEK293T: 4×10^5 cells/well,

UE7T-9: 2×10^5 cells/well). After 48 h of incubation, each cell culture medium was collected

and analyzed. The secreted CXCL12 level was assessed using a Human CXCL12/SDF-1 Alpha

Quantikine ELISA Kit (R&D Systems), following the manufacturer’s protocol.

Legend of Figure 4 (Line 799-811)

f, g CXCL12 secretion was significantly up-regulated in HEK293T (f) and UE7T-9 (g) cells of

C9orf89-CKO, *MAGI2*-CKO, *MLPH*-CKO, and *RHBDD2*-CKO. h, i In the co-culture

experiment of U937 with *RHBDD2*-CKO HEK293T (h) or *RHBDD2*-CKO UE7T-9 (i), pre-treatment with CXCL12-neutralizing antibodies increased cleaved caspase-3-positive cells under cytarabine exposure for 48 h, which was similar to the effect observed when the cells were co-cultured with control HEK293T or UE7T-9. j, k Evaluation of phospho-Akt downstream of CXCL12 in GFP-positive U937 under mono- or co-culture conditions. Phospho-Akt was increased in U937 co-cultured with *RHBDD2*-CKO HEK293T (j) and *RHBDD2*-CKO UE7T-9 (k) for 48 h. Experiments were performed with biological triplication (a, d, e, g–k) or quintuplication (f) in three independent repeats. Data are represented as mean \pm SD. Statistical significance values were calculated by performing one-way ANOVA with Dunnett’s test (a, d–g) or one-way ANOVA with Bonferroni’s test (h–k). * $p < 0.05$; ** $p < 0.01$; *** $p < 0.001$; and NS: non-significant.

6. Do the different knockouts in HEK293 also show resistance to higher doses of the drug in monocultures without tumors?

Response: Thank you for your pertinent question. We added Supplementary Figure 9, which includes killing curves of HEK293T cells with cytarabine exposure. Cell viabilities of HEK293T cells of *C9orf89*-CKO, *MAGI2*-CKO, *MLPH*-CKO, and *RHBDD2*-CKO, and Non-target, under cytarabine exposure were assessed using MTS assay. No significant differences in viabilities were observed in each concentration of cytarabine.

Supplementary Figure 9.

Supplementary Figure 9.

Cell viabilities of HEK293T-CKO clones under cytarabine exposure. Dose–response curves for HEK293T cells of *C9orf89*-CKO, *MAGI2*-CKO, *MLPH*-CKO, and *RHBDD2*-CKO, and

Non-target treated with cytarabine for 48 h with biological triplication. IC50s to cytarabine are shown. Data are represented as mean \pm SD. Statistical significance values were calculated by performing one-way ANOVA with Dunnett's test. No significant differences in viabilities were observed for any concentration of cytarabine.

The revised part is highlighted in the manuscript as follows:

Results (Line 195-198)

Furthermore, no significant differences in the viabilities of HEK293T cells of *C9orf89*-CKO, *MAGI2*-CKO, *MLPH*-CKO, *RHBDD2*-CKO, and Non-target were observed under cytarabine exposure (Supplementary Fig. 9).

7. What do the x-axis in figure 4f represent? Please provide the scatter plots by flow. What is the gating strategy?

Response: We apologize for the confusion caused. The x-axis in Figure 4f represents, from left to right, U937 cells mono-culture system, U937 with HEK293T-Non-target cells co-culture system, and U937 with HEK293T-*RHBDD2*-CKO cells co-culture system. The y-axis in Figure 4f represents the expression of phospho-Akt in U937-GFP-positive cells in each culture system.

In the analysis of phospho-Akt of U937 co-cultured with HEK293T or UE7T-9, U937 was labeled with GFP to evaluate phospho-Akt in U937 cells only. The expression of phospho-Akt of U937 was evaluated using the BD FACSCanto II. The cells were gated based on FSC-A and SSC-A channels to exclude debris and dead cells, and they were further gated based on the GFP-A channel to exclude HEK293T or UE7T-9. The mean fluorescence of PE in the GFP-positive population was evaluated as the expression of phospho-Akt in U937-GFP-positive cells.

Scatter plots for the gating strategy as follows are newly provided in Supplementary Figure 12. Additionally, Figure 4f is revised as Figure 4j in the manuscript. The same strategy is applied to Figure 4k, which mentions experiments with UE7T-9-U937 co-culture systems.

Supplementary Figure 12.

Supplementary Figure 12. Scatter plots of the gating strategy for the analysis of phospho-Akt in the co-culture experiments in U937-GFP-positive cells with HEK293T cells.

Expression of phospho-Akt in U937-GFP-positive cells in the present culture system was evaluated by FACSCanto II. The cells were gated based on FSC-A and SSC-A channels to exclude debris and dead cells, and they were further gated based on the GFP-A channel to exclude HEK293T. The mean fluorescence of PE in the GFP-positive population was evaluated as the expression of phospho-Akt in U937-GFP-positive cells.

The following content has been added and is highlighted in the manuscript:

Methods (Line 526-528)

The mean fluorescence of PE in the GFP-positive population was evaluated as the expression of phospho-Akt in U937-GFP-positive cells. Gating strategy is provided in Supplementary Fig. 12.

8. Please provide a tab delimited summary file for the gene level RNAseq data

Response: RNA-seq data including a tab-delimited summary file for the gene-level RNA-seq data have been deposited in the Gene Expression Omnibus (GEO) under access number GSE203256 (<https://www.ncbi.nlm.nih.gov/gds/?term=GSE203256>).

9. Please provide cell numbers for figures 2b,c, 3

Response: As noted in the Data Availability section of the manuscript, the cell numbers for Figures 2b,c and 3 have been deposited at figshare.com (DOI: 10.6084/m9.figshare.22492372) (https://figshare.com/articles/dataset/Revised_Raw_data_Indirect_CRISPR_screening_with_photoconversion_revealed_key_factors_of_drug_resistance_with_cell_cell_interactions/22492372). Please refer to the files “Values for the graph in Figure 2b,” “Values for the graph in Figure 2c,” “Values for the graph in Figure 3a,” and “Values for the graph in Figure 3b” on the above Figshare link.

Reviewer #2 (Remarks to the Author):

The manuscript by Sugita and coauthors reports an indirect CRISPR screening system with photoconvertible fluorescent protein Dendra2 to screen for key factors responsible for drug resistance with cell-cell interactions. The authors first applied this indirect CRISPR screening approach in HEK293T-U937 co-culture system and found 39 candidate genes with drug resistance functions in supporting cells. Then they validated these candidate genes in several different co-culture systems, investigated their biological features, and found a new axis linking RHBDD2, CXCL12, and PI3k-Akt-mTOR which is responsible for anticancer drug resistance. Finally, they analyzed the expression level of four important candidate genes in clinical pancreatic cancer samples to show their significance as prognostic factors. Overall, this manuscript is well-written with clear rationale and detailed methods for reproducing the work. The indirect CRISPR screening system is novel with good potential to be applied for screening in other cell-cell interaction systems. The discovery of RHBDD2-CXCL12 axis is also new and interesting. The reviewer only has several concerns.

Thank you for your thoughtful analysis. We appreciate that you found the study *well-written with clear rationale and detailed methods* and that you are interested in the new axis responsible for anticancer drug resistance. We also thank you generally for your time and valuable feedback.

Concerns:

1. In Fig. 2 and Supplementary Figure 3, the viability of U937 cells in the co-culture system was assessed by quantifying GFP-positive cells as viable U937 cells. The reviewer is concerned if the GFP-positive cells can correctly represent viable cells.

Response: Thank you for your thoughtful consideration. We added new data on GFP expression in the Propidium iodide (PI)-positive cells or Annexin V-positive cells in the present co-culture system (Supplementary Figure 2d, e). In the co-culture system with HEK293T cells, GFP-positive U937 cells were exposed to cytarabine. After 48 h with exposure to cytarabine, PI or Annexin V was evaluated. PI-positive or Annexin V-positive cells increased with cytarabine exposure, however, in the population of the GFP-positive cells (No gated on FSC and SSC), few PI-positive or Annexin V-positive cells were observed and no significant increase was identified at any cytarabine concentration. Therefore, almost all PI-positive or Annexin V-positive cells do not express GFP, and concluded that GFP-positive U937 cells represent living U937 cells and this system is useful in the present screening.

Supplementary Figure 2.

Supplementary Figure 2.

(d, e) GFP-positive cells represent viable U937 cells. In the co-culture system with HEK293T cells, GFP-positive U937 cells were exposed to cytarabine. After 48 h of exposure to cytarabine, Propidium iodide (PI) (d) or Annexin V (e) was evaluated. Under the non-gated condition, PI-positive (d) or Annexin V-positive (e) cells increased with cytarabine exposure. However, under the GFP-gated condition, few PI (d) or Annexin V (e) positive cells were observed at any cytarabine concentration. So, almost all PI-positive or Annexin V-positive cells do not express GFP, and therefore, GFP-positive U937 cells represent living U937 cells. The experiment was performed with biological triplication in the two independent experiments. Data are represented as mean \pm SD. Statistical significance values were calculated by performing one-way ANOVA with Dunnett's test (d, e). In the analysis of PI or Annexin V (d, e), the control was "cytarabine 0 μ M" of each gating condition. ** p < 0.01, *** p < 0.001, and NS: non-significant.

The revised part is highlighted in the manuscript as follows:

Results (Line 130-132)

In the present co-culture system, GFP-positive U937 cells represent viable U937 cells (Supplementary Fig. 2d, e).

2. There are 39 candidate genes identified by screening. However, only 11 genes were successfully validated with drug resistance functions (Fig. 2). How do you explain why the other 28 candidate genes were identified by screening but not successfully validated? Does that mean the indirect CRISPR screening assay has a high false positive ratio?

Response: Thank you for your consideration. There could be many reasons why some candidate genes were identified by screening but not successfully validated. Generally, this will have been caused by the technical limitations of random screening and the original systems used in the present study.

In the screening experiment, HEK293T cells, in close proximity to viable colonies identified under microscopic observation, were manually illuminated by the laser. We speculate that the false-positive rate in labeling candidate cells was caused by the inevitable exposure of the laser beam to non-objective cells surrounding the objective candidate cells.

Additionally, in the co-culture validation experiments, the concentration of cytarabine was 5 μM , higher than the concentration used in screening, 3 μM , to select candidate genes of stronger phenotypes inducing anticancer drug resistance.

Of course, off-target effects of gRNAs were also potentially responsible for false-positive candidates.

To overcome the problem of false positives, repeated validation experiments were conducted using multiple cell types in the present study. We are currently considering methods to improve the system to isolate objective candidate cells accurately and efficiently.

3. There is no clear description and citation of the CRISPR knockout library used for the indirect CRISPR screening. Please add this information in the manuscript.

Response: Thank you for pointing out this missing information. It was provided in Supplementary Table 1 “Summary of Plasmids,” but a description of the library was missing from the main text. We used the Human CRISPR Knockout Pooled Library (GeCKO v2) (Addgene, MA, USA, #100000049). Accordingly, an additional citation has been added as No. 36 in the manuscript and to the References section as follows:

The revised parts are highlighted in the manuscript as follows:

Methods (Line 387-388)

The Human CRISPR Knockout Pooled Library A (Addgene, MA, USA, #100000049) in lentiGuide-Puro³⁶ was transduced into Cas9-Dendra2-expressing HEK293T cells and then treated with puromycin (1 $\mu\text{g}/\text{ml}$, InvivoGen) for 2 weeks.

References (Line 691-692)

36. Sanjana, N. E., Shalem, O. & Zhang, F. Improved vectors and genome-wide libraries for CRISPR screening. *Nat. Methods* 8, 783–784 (2014).

4. For line 120-121, please describe how the conditions for drug selection was tightened.

Response: We appreciate this comment. The text under editing was left in accidentally. We have replaced the incorrect sentence with the following, which now complements the sentences before and after.

Results (Line 119-121)

Collected mutated HEK293T cells were then expanded and the mutation was analyzed using TA cloning, after which 39 candidate genes were obtained (Fig. 1b, Step 7, Table 1).

5. Please include a calibration bar in Supplementary Figure 1 to show the relative intensity of the red channel.

Response: A calibration bar showing the relative intensity of the red has been added to Supplementary Figure 1.

Supplementary Figure 1.

6. It's better to include a diagram to describe the newly discovered axis responsible for anticancer drug resistance in the main figure to help the readers to understand.

Response: We appreciate this comment. Figure 5d was inserted to explain Line 308-310 in the Discussion and demonstrate the newly discovered axis.

Figure 5d

Figure 5d. (Line 826-828)

d Schematic representation of RHBDD2-CXCL12 axis in drug resistance with cell-cell interactions.

7. For line 103, please include a comma after “As we expected”.

Response: We appreciate this comment and corrected the error in question.

We believe we have successfully addressed both the reviewers’ questions and concerns. Thank you again for your time and valuable feedback.

REVIEWERS' COMMENTS:

Reviewer #1 (Remarks to the Author):

I believe the authors addressed all my comments and I think the paper should be accepted.

Reviewer #2 (Remarks to the Author):

The authors addressed most of the reviewer's concerns properly. The reviewer only has two suggestions here.

1. For concern 2, it's better to include the discussion about why many candidate genes identified by screening were not successfully validated into the discussion part of the manuscript. It helps the readers to understand the reported method thoroughly and also pay attention to this limitation when they try to apply this method.

2. For concern 5, the reviewer means to add a calibration bar to each of the two red channels. Otherwise, it's hard to compare the intensity of the red channel before and after 405 nm laser illumination.

Manuscript ID: COMMSBIO-22-3524A

Indirect CRISPR screening with photoconversion revealed key factors of drug resistance with cell-cell interactions

We would like to thank the reviewers for their thoughtful and thorough analysis of our manuscript. We have endeavored to address all of the issues they presented. The response comments are in **bold and blue**.

REVIEWERS' COMMENTS:

Reviewer #1 (Remarks to the Author):

I believe the authors addressed all my comments and I think the paper should be accepted.

Thank you for your time and valuable feedback.

Reviewer #2 (Remarks to the Author):

The authors addressed most of the reviewer's concerns properly. The reviewer only has two suggestions here.

Thank you for your thoughtful analysis. We also thank you generally for your time and valuable feedback.

1. For concern 2, it's better to include the discussion about why many candidate genes identified by screening were not successfully validated into the discussion part of the manuscript. It helps the readers to understand the reported method thoroughly and also pay attention to this limitation when they try to apply this method.

We appreciate this comment. We have added the following text in the discussion part of the manuscript.

Discussion (Line 298-312)

In the present study, there are 39 candidate genes identified by screening. However, only 11 genes were successfully validated with drug resistance functions. There could be many reasons why some candidate genes were identified by screening but not successfully validated.

Generally, this will have been caused by the technical limitations of random screening and the original systems used in the present study. In the screening experiment, HEK293T cells, in close proximity to viable colonies identified under microscopic observation, were manually illuminated by the laser. We speculate that the false-positive rate in labeling candidate cells was caused by the inevitable exposure of the laser beam to non-objective cells surrounding the

objective candidate cells. Additionally, in the co-culture validation experiments, the concentration of cytarabine was 5 μM , higher than the concentration used in screening, 3 μM , to select candidate genes of stronger phenotypes inducing anticancer drug resistance. Of course, off-target effects of gRNAs were also potentially responsible for false-positive candidates. To overcome the problem of false positives, repeated validation experiments were conducted using multiple cell types in the present study. We are currently considering methods to improve the system to isolate objective candidate cells accurately and efficiently.

2. For concern 5, the reviewer means to add a calibration bar to each of the two red channels. Otherwise, it's hard to compare the intensity of the red channel before and after 405 nm laser illumination.

Thank you for pointing out. We add a calibration bar to each of the two red channels, before and after photoconversion, to Supplementary Figure 1.

Supplementary Figure 1.

We believe we have successfully addressed all of the reviewers' concerns.

Thank you again for your time and valuable feedback.